# The LIM protein complex establishes a retinal circuitry of visual adaptation by regulating Pax6 α-enhancer activity

Yeha Kim[1], Soyeon Lim[1†], Taejeong Ha[1†], You-Hyang Song[1], Young-In Sohn[1], Dae-Jin Park[2], Sun-Sook Paik[3], Joo-ri Kim-Kaneyama[4], Mi-Ryoung Song[5], Amanda Leung[6], Edward M Levine[6], In-Beom Kim[3], Yong Sook Goo[2], Seung-Hee Lee[1], Kyung Hwa Kang[7], Jin Woo Kim[1]*

[1]Department of Biological Sciences, Korea Advanced Institute of Science and Technology (KAIST), Daejeon, South Korea; [2]Department of Physiology, Chungbuk National University School of Medicine, Cheongju, South Korea; [3]Department of Anatomy, College of Medicine, The Catholic University of Korea, Seoul, South Korea; [4]Department of Biochemistry, Showa University School of Medicine, Tokyo, Japan; [5]Department of Life Sciences, Gwangju Institute of Science and Technology (GIST), Gwangju, South Korea; [6]Department of Ophthalmology and Visual Sciences, Vanderbilt University, Nashville, United States; [7]KAIST Institute of BioCentury, Daejeon, South Korea

*For correspondence:
jinwookim@kaist.ac.kr

[†]These authors contributed equally to this work

Competing interests: The authors declare that no competing interests exist.

**Abstract** The visual responses of vertebrates are sensitive to the overall composition of retinal interneurons including amacrine cells, which tune the activity of the retinal circuitry. The expression of *Paired-homeobox 6 (PAX6)* is regulated by multiple cis-DNA elements including the intronic α-enhancer, which is active in GABAergic amacrine cell subsets. Here, we report that the transforming growth factor ß1-induced transcript 1 protein (Tgfb1i1) interacts with the LIM domain transcription factors Lhx3 and Isl1 to inhibit the α-enhancer in the post-natal mouse retina. $Tgfb1i1^{-/-}$ mice show elevated α-enhancer activity leading to overproduction of Pax6ΔPD isoform that supports the GABAergic amacrine cell fate maintenance. Consequently, the $Tgfb1i1^{-/-}$ mouse retinas show a sustained light response, which becomes more transient in mice with the auto-stimulation-defective $Pax6^{\Delta PBS/\Delta PBS}$ mutation. Together, we show the antagonistic regulation of the α-enhancer activity by Pax6 and the LIM protein complex is necessary for the establishment of an inner retinal circuitry, which controls visual adaptation.

## Introduction

The retina is a primary sensory tissue that receives light stimulus and converts it into electrical signals, which are then sent to the brain for further processing. After light detection by rod and cone photoreceptors, the first step in visual processing occurs in bipolar cells that are either stimulated or inhibited by light-absorbed photoreceptors (*Masland, 2012*). The activities of bipolar cells are then tuned by horizontal cells while they receive visual input from the photoreceptors and by amacrine cells while they deliver the signals to retinal ganglion cells (RGCs) (*Hoon et al., 2014*; *Masland, 2012*). The amacrine cells do not simply convey the signals from bipolar cells, but they also invert the signals by releasing inhibitory neurotransmitters such as γ-aminobutyric acid (GABA) and glycine. Therefore, even subtle changes in the composition and connectivity of amacrine cell subsets might alter the output of the retina, modifying the visual information sent to the brain.

**eLife digest** The retina is a light-sensitive layer of tissue that lines the inside of the eye. This tissue is highly organized and comprises a variety of different nerve cells, including amacrine cells. Together, these cells process incoming light and then trigger electrical signals that travel to the brain, where they are translated into an image. Changes in the nerve cell composition of the retina, or in how the cells connect to each other, can alter the visual information that travels to the brain.

The nerve cells of the retina are formed before a young animal opens its eyes for the first time. Proteins called transcription factors – which regulate the expression of genes – tightly control how the retina develops. For example, a transcription factor called Pax6 drives the development of amacrine cells. Several other transcription factors control the production of Pax6 by binding to a section of DNA known as the "α-enhancer". However, it is not clear how regulating Pax6 production influences the development of specific sets of amacrine cells.

Kim et al. reveal that a protein known as Tgfb1i1 interacts with two transcription factors to form a "complex" that binds to the α-enhancer and blocks the production of a particular form of Pax6. In experiments performed in mice, the loss of Tgfb1i1 led to increased production of this form of Pax6, which resulted in the retina containing more of a certain type of amacrine cell that produce a molecule called GABA. Mice lacking Tgfb1i1 show a stronger response to light and are therefore comparable to people who are too sensitive to light. On the other hand, mice with a missing a section of the α-enhancer DNA have fewer amacrine cells releasing GABA and become less sensitive to light and are comparable to people who have difficulty detecting weaker light signals.

The findings of Kim et al. suggest that an individual's sensitivity to light is related, at least in part, to the mixture of amacrine cells found in their retina, which is determined by certain transcription factors that target the α-enhancer.

The neurons of the vertebrate retina develop in an ordered fashion from multipotent retinal progenitor cells (RPCs) (*Cepko, 2014*). A number of transcription factors with precise temporal and spatial expression patterns control the composition of retinal neurons via the hierarchical and reciprocal regulation of other transcription factor expression (*Zagozewski et al., 2014*). Thus, the alterations of transcription factors that specify retinal neuron subtypes should modify visual output of mature retina. Those transcription factors include Pax6 in amacrine cells (*Marquardt et al., 2001*), Vsx2 in bipolar cells (*Liu et al., 1994*), Otx2 in bipolar cells and photoreceptors (*Koike et al., 2007*; *Nishida et al., 2003*), and Lhx2 and Sox2 in Müller glia and certain amacrine subtypes (*de Melo et al., 2012*; *Gordon et al., 2013*; *Lin et al., 2009*). These transcription factors are not only expressed in the earlier optic structures to play critical roles in the eye and brain development (*Danno et al., 2008*; *Glaser et al., 1994*; *Yun et al., 2009*), but also in the mature retinal neurons to support the survival and functions of the neurons (*de Melo et al., 2012*; *Kim et al., 2015*). However, the mechanisms underlying the recurrent expression of transcription factors in the retinal lineage are still largely unknown.

Pax6 is one of the earliest transcription factors expressed in the eye field, and as such, it is considered as a master regulator of eye development (*Ashery-Padan and Gruss, 2001*; *Hanson and Van Heyningen, 1995*). Pax6 contains two DNA-binding domains—a paired domain (PD) and a homeodomain (HD)—linked via a glycine-rich domain, and activates target gene transcription through its C-terminal proline-, serine-, and threonine-rich (PST) domain (*Epstein et al., 1994*; *Xu et al., 1999a*). Multiple *cis*-regulatory elements govern *Pax6* expression in various mouse tissues (*Kammandel et al., 1999*; *Xu et al., 1999b*). The 'α-enhancer', located within intron 4 of the *Pax6* gene, is active in the retina from embryo to adult (*Kammandel et al., 1999*; *Marquardt et al., 2001*; *Plaza et al., 1995*). This retina-specific enhancer activity sustains in RPCs in the peripheral retina of the embryos and regulates neuronal differentiation in a context-dependent manner (*Marquardt et al., 2001*). In the mature eye, the α-enhancer is active in cells of the ciliary body and amacrine cells of the retina (*Marquardt et al., 2001*).

The α-enhancer contains multiple binding sites for transcription factors, including the auto-stimulatory Pax6 (*Kammandel et al., 1999*), the stimulatory Msx1 (*Kammandel et al., 1999*) and Pou4f2

(*Plaza et al., 1999*), and the inhibitory Pax2 (*Kammandel et al., 1999*; *Schwarz et al., 2000*) and Vax1 (*Mui et al., 2005*). Although the inhibition of α-enhancer activity by Vax1 has been shown to be crucial for the development of the retina-optic stalk border (*Mui et al., 2005*), the roles the other transcription factors that bind the α-enhancer in the retina remain unclear. In this study, we show that regulation of *Pax6* expression through the α-enhancer fine tunes amacrine cell subtype composition, and consequently, the visual output of the retina.

## Results

### Identification of Lhx3 and Tgfb1i1 as Pax6 α-enhancer binding proteins in mouse retina

According to DNase footprinting (DF) results, the *Pax6* α-enhancer contains four retina-specific transcription factor-binding sites called DF1–4 (*Plaza et al., 1995*). It also contains an auto-regulatory Pax6 binding sequence (PBS; *Figure 1A*). The AT-rich region designated DF4 recruits both positive and negative regulators expressed in the optic vesicle and embryonic retina (*Lakowski et al., 2007*; *Mui et al., 2005*; *Plaza et al., 1999*; *Schwarz et al., 2000*). Still, the transcription factors responsible for regulating α-enhancer activity in the post-natal retina are not yet known.

In a proteomic screen for DF4-binding proteins in R28 rat RPCs, we identified Lhx3 (LIM domain homeobox 3) and Hic-5 (hydrogen peroxide induced clone 5)/Tgfb1i1 (tumor growth factor-$\beta$1 induced transcript one protein)/Ara55 (androgen receptor-associated protein 55) as potential candidates (*Figure 1B*; see Materials and methods for details). These proteins share the LIM (LIN-11, Isl1, or MEC-3) protein-protein interaction domain (*Karlsson et al., 1990*; *Way and Chalfie, 1988*). In addition, Lhx3 contains a homeodomain and acts as a transcription factor (*Bridwell et al., 2001*; *Roberson et al., 1994*). Tgfb1i1 has four leucine-rich domains (LDs), which mediate interactions with other LD-containing protein, and four LIM domains, which mediate both self-oligomerization and interactions with other LIM domain-containing proteins (*Mori et al., 2006*; *Nishiya et al., 1999*).

Lhx3 is absent from the embryonic mouse retina, but is expressed in bipolar cells beginning around the first post-natal week (*Figure 1C*, top; *Figure 1—figure supplement 1A*) (*Balasubramanian et al., 2014*). Tgfb1i1 is expressed in most of post-natal retina, but is absent from the embryonic and adult mouse retinas (*Figure 1C*, bottom; *Figure 1—figure supplement 1B*). We also noticed Lhx3- and Tgfb1i1-expressing cells in P8 retinas show no *Pax6* α-enhancer activity (*Figure 1C*), as visualized by an GFP reporter in *Pax6 α-enhancer::Cre-IRES-GFP* (*P6α-CreiGFP*) mice (*Marquardt et al., 2001*). This suggests a potential negative relationship between these LIM domain proteins and the α-enhancer activity.

To validate our screening results, we further examined the binding of those LIM-domain containing proteins in P7 retinal nuclear extracts to DF4 sequence, and found Lhx3 and Tgfb1i1 in these extracts bind wild-type DF4 dsDNA (DF4-WT) but not mutant DF4 dsDNA (DF4-MUT) in which the 5'-ATTA-3' homeodomain target sequence is replaced with 5'-CGGC-3' (*Figure 1—figure supplement 2A*). Not only the endogenous Lhx3 but also in vitro-translated Lhx3 specifically binds the DF4 oligomer (*Figure 1—figure supplement 2B*). In vitro-translated Tgfb1i1, however, lacks a DNA-binding motif, and so does not bind the DF4 oligomer (*Figure 1—figure supplement 2B*). This suggests Tgfb1i1 binds the α-enhancer indirectly, possibly via an interaction with another DF4-binding protein like Lhx3.

To determine whether Lhx3 and Tgfb1i1 bind the α-enhancer in vivo, we performed a chromatin immunoprecipitation (ChIP) analysis using rabbit polyclonal antibodies raised against Lhx3 or Tgfb1i1. We checked the ChIP DNA fragments isolated from P7 retinas for two mouse *Pax6* gene sequences located in the ectodermal enhancer of the 5'-UTR and the α-enhancer of intron 4 using PCR (*Figure 1D*) and quantitative PCR (qPCR; *Figure 1E*). Since both of these enhancer elements include auto-regulatory Pax6 binding sequences (*Aota et al., 2003*; *Kammandel et al., 1999*), we used ChIP DNA fragments obtained with anti-Pax6 rabbit IgG (α-Pax6) as a positive control and those obtained with pre-immune rabbit IgG (RbIgG) as a negative control. We found that, in the mouse retina, Lhx3 and Tgfb1i1 interact specifically with the α-enhancer but not the ectodermal enhancer (*Figure 1D,E*).

As other LIM domain-containing transcription factors can target the same DNA sequences as Lhx3 (*Gehring et al., 1994*), we also determined whether other LIM domain transcription factors

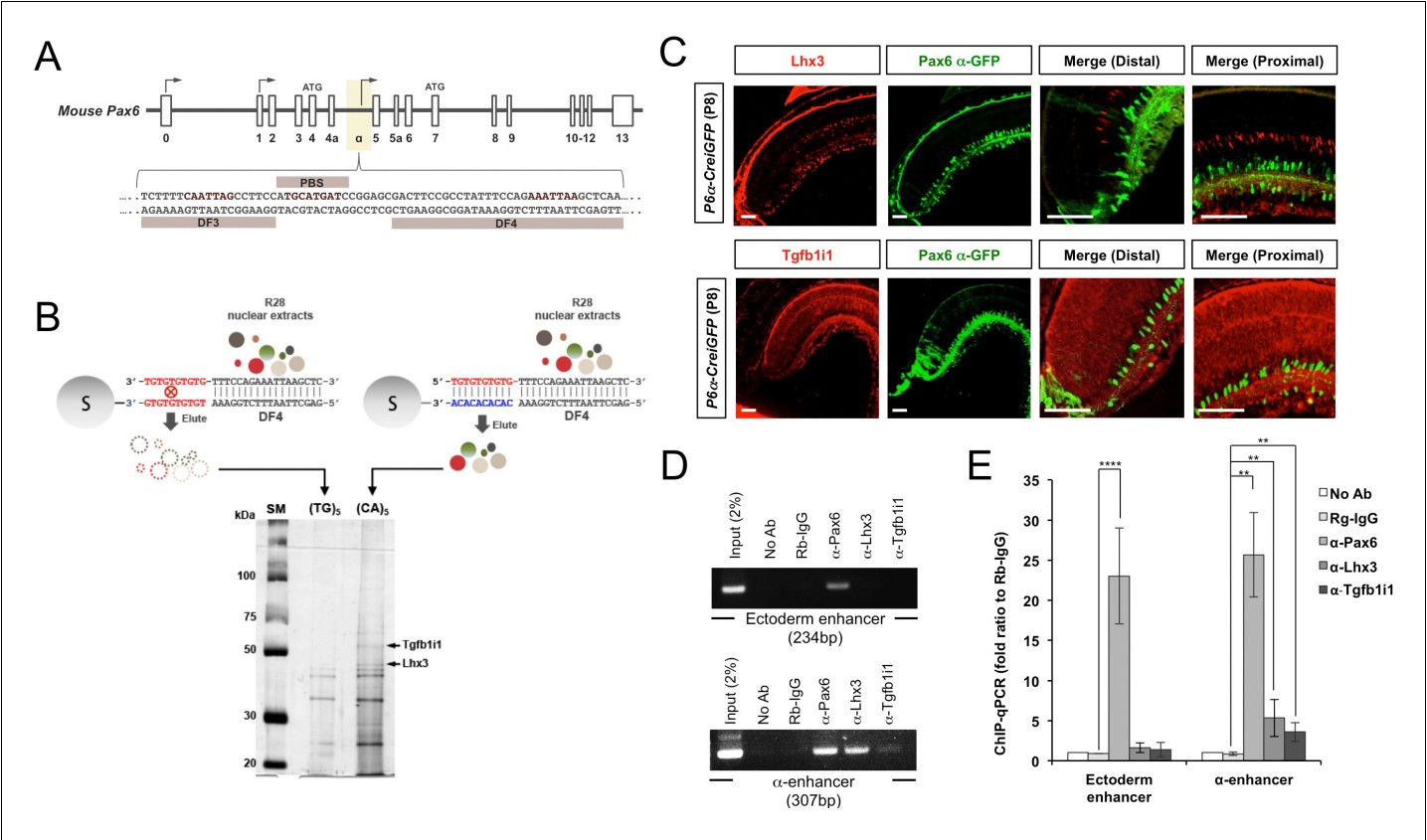

**Figure 1.** Identification of Lhx3 and Tgfb1i1 as Pax6 α-enhancer binding proteins. (**A**) (Top) The genomic structure of the mouse *Pax6* gene. Exons are shown as boxes, and arrows denote transcription initiation sites. (Bottom) The DF3, PBS, and DF4 sequences in the retina-specific α-enhancer are indicated with their core homeodomain (HD) and paired domain (PD) binding sites colored red. (**B**) Nuclear extracts from R28 rat retinal precursor cells were incubated with DF4 dsDNA oligomers with single-stranded 5'-(GT)$_5$-3' overhangs. DF4 oligomer-protein complexes were then added to Sepharose 6B columns conjugated with single-stranded DNA (ssDNA) of 5'-(CA)$_5$-3', which is complementary to the single-stranded overhang sequence of the oligomer, or 5'-(TG)$_5$-3' non-specific binding control. Proteins bound to the ssDNA column were eluted for SDS-PAGE and detected by silver staining. Protein bands specifically enriched in the (CA)$_5$ column were then eluted from the gel and digested for mass spectrometric identification. This analysis identified the two bands marked by arrows as Lhx3 and Tgfb1i1. (**C**) Lhx3 and Tgfb1i1 expression in post-natal day 8 (P8) *Pax6 α-enhancer::Cre-IRES-GFP* (*P6α-CreiGFP*) mouse retinas stained with rabbit anti-Lhx3 (top) and anti-Tgfb1i1 (bottom) antibodies (red). These were also co-stained with a chick anti-GFP antibody (green). Scale bars, 100 μm. (**D**) DNA fragments bound to Pax6, Lhx3, and Tgfb1i1 in P7 mouse retinas were isolated by chromatin immunoprecipitation (ChIP) using rabbit polyclonal antibodies against each protein. The relative enrichment of each protein on the ectoderm enhancer and the α-enhancer of *Pax6* gene was determined by PCR amplification of each enhancer sequence from the ChIP DNA fragments. (**E**) qPCR threshold cycle (Ct) values for each ChIP sample were compared to those of a protein-A bead only sample to obtain relative expression ($2^{-ΔCt}$). The graph shows the ratio of $2^{-ΔCt}$ values for each sample to those of a pre-immune rabbit IgG (Rb-IgG) ChIP sample. Error bars indicate standard deviations (STD, n = 5).

The following figure supplements are available for figure 1:

**Figure supplement 1.** Lhx3 and Tgfb1i1 expression in embryonic and mature mouse retinas.

**Figure supplement 2.** Binding abilities of Lhx3 and Tgfb1i1 to *Pax6* α-enhancer sequence.

**Figure supplement 3.** Relationship between LIM domain transcription factor expression and *Pax6* α-enhancer activity in mouse retina.

expressed in the post-natal retina, such as Islet-1 (Isl1) and Lhx2 (*Balasubramanian et al., 2014*) (*Figure 1—figure supplement 3A*), also can bind the Pax6 α-enhancer DF4 sequence. We found Lhx2, but not Isl1, shows specific binding to the DF4 sequence (*Figure 1—figure supplement 3C*). Isl1 instead binds DF3, which contains the predicted Isl1 binding sequence 5'-CATTAG-3' (*Lee et al., 2008*; *Leonard et al., 1992*) (*Figure 1—figure supplement 3D*). Conversely, the DF4-recognizing

LIM transcription factors Lhx2 and Lhx3 do not bind the DF3 sequence (*Figure 1—figure supplement 3D*). Collectively, these results suggest Tgfb1i1 binds the α-enhancer indirectly, possibly via an interaction with these LIM domain transcription factors.

## Lhx3 and Isl1 inhibit Pax6 α-enhancer activity in a Tgfb1i1-dependent manner

Lhx3 is expressed in cone bipolar cells but not in amacrine cells, including the Pax6 α-GFP-positive subpopulation (*Figure 1C*; *Figure 1—figure supplement 3A,B*) (*Balasubramanian et al., 2014*). On the contrary, Lhx2 is expressed primarily in Müller glia (*Balasubramanian et al., 2014*) but also in amacrine cells, including those with α-enhancer activity (*Figure 1—figure supplement 3A,B*). Lhx9 is also expressed in amacrine cells (*Balasubramanian et al., 2014*), about 60% of which show *Pax6 α*-enhancer activity (*Figure 1—figure supplement 3A,B*). Both Lhx2 and Lhx9 activate the *Pax6 α*-luciferase reporter in a dose-dependent manner (*Figure 2A,B*). In contrast, Lhx3 and Lhx4 do not affect α-enhancer activity alone (*Figure 2A*), but they antagonize Pax6-induced activation of the α-enhancer (*Figure 2B*). Isl1 is expressed in ON bipolar cells and cholinergic amacrine cells (*Elshatory et al., 2007*; *Galli-Resta et al., 1997*; *Haverkamp et al., 2003*), but not in *Pax6 α*-enhancer-active amacrine cells (*Figure 1—figure supplement 3A,B*). Isl1 does not affect *Pax6 α*-

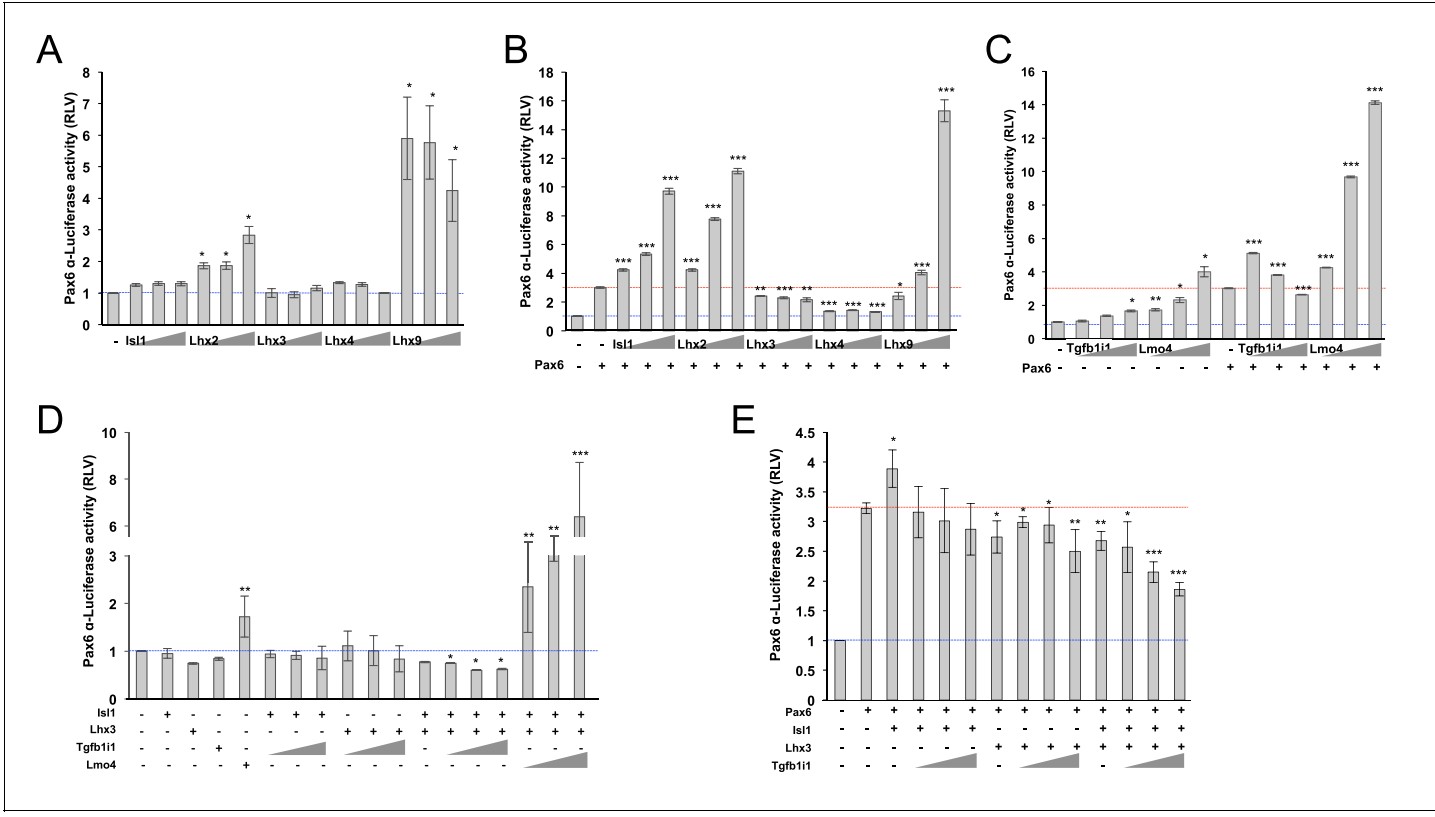

**Figure 2.** Lhx3 and Isl1 inhibit Pax6 α-enhancer activity in a Tgfb1i1-sensitive manner. (A) The effects of LIM domain transcription factors on *Pax6 α*-enhancer activity were measured with a *Pax6 α*-enhancer luciferase reporter in HEK293T cells. These cells were co-transfected with DNA constructs encoding cDNAs of the indicated genes as well as the *Pax6 α*-luciferase reporter (0.2 μg). The triangles denote increasing doses of the indicated constructs (0.5 μg, 1 μg, and 2 μg). The relative luciferase activity of each sample was normalized to co-expressed β-galactosidase activity. (B) The effects of LIM domain transcription factors on Pax6-induced activation of the α-enhancer were also examined in the cells transfected with same DNA constructs used in (A) plus Pax6 construct (0.5 μg). (C) Regulatory effects of Tgfb1i1 and Lmo4 on *Pax6 α*-enhancer activity were also examined in the transfected cells as described in (A) and (B). (D and E) Cooperative effects of Isl1, Lhx3, and Tgfb1i1 on *Pax6 α*-enhancer activity were examined with the indicated combinations. (A – E) The blue lines indicated relative luciferase activity in samples expressing only the luciferase reporter, while red lines indicate activity of samples expressing the reporter with Pax6. The values on the Y-axes are averages. Error bars indicate STD (n > 5); *p<0.05, **p<0.01, ***p<0.001.

enhancer activity alone, but it does activate the enhancer in the presence of Pax6 (*Figure 2A,B*). Together, these results suggest LIM domain transcription factors in the mouse retina can be categorized based on how they affect *Pax6* α-enhancer—some are stimulatory (i.e., Lhx2 and Lhx9), some are inhibitory (i.e., Lhx3 and Lhx4), and some are context-sensitive (i.e., Isl1).

Tgfb1i1, although it is unable to bind the α-enhancer directly (*Figure 1—figure supplement 2B*), inhibits Pax6-induced α-enhancer activity upon overexpression (*Figure 2C*). This inhibition of the α-enhancer is even more significant when Tgfb1i1 is co-expressed with both Lhx3 and Isl1 (*Figure 2D, E*). We hypothesized that multiple LIM domains of Tgfb1i1 allow it to form a multi-protein complex that blocks α-enhancer-dependent gene expression. To assess this, we co-expressed these LIM domain transcription factors with Lmo4 (LIM-domain-only 4), which prevents Isl1 and Lhx3 from interacting with one another or with other LIM domain-containing proteins (*Thaler et al., 2002*). Lmo4 alone caused a dose-dependent increase in α-enhancer activity and potentiated Pax6-induced activation of the α-enhancer (*Figure 2C*). In the presence of Lmo4, Lhx3 and Isl1 cannot inhibit the α-enhancer (*Figure 2D*). Thus, Tgfb1i1 and Lmo4 appear to oppositely regulate α-enhancer activity by antagonistically modulating the formation of the LIM domain transcription factor complex.

## Pax6 and Tgfb1i1 competitively bind Isl1 to antagonistically regulate the α-enhancer

We next used co-immunoprecipitation to determine whether Isl1, Lhx3, and Tgfb1i1 form a LIM protein complex in P7 mouse retina. We were able to detect Isl1 in complexes recovered using Lhx3 and Tgfb1i1, which are also capable of precipitating one another (*Figure 3A*). This suggests these three proteins may exist as a complex in the retina. To further examine the molecular nature of this LIM protein complex, we used combinatorial transfections of constructs encoding Lhx3, Isl1, and Tgfb1i1 into human embryonic kidney 293T (HEK293T) cells. The results of these transfections are summarized in *Figure 3—source data 1*.

In the co-immunoprecipitation experiments, we found that Isl1 binds to Lhx3 with its homeodomain (HD) and/or LIM binding domain (LBD), as reported previously (*Thaler et al., 2002*), whereas it interacts with Tgfb1i1 with its LIM domain(s) (*Figure 3B* [left column]; *Figure 3—figure supplement 1B,C*). Lhx3 binds to Tgfb1i1 and Isl1 via its LIM domain(s) (*Figure 3B* [center column]; *Figure 3—figure supplement 1E,F*). Tgfb1i1 also uses LIM domain(s) to interact with Isl1 and Lhx3 (*Figure 3B* [right columns]; *Figure 3—figure supplement 1H,I*). We further tested whether Tgfb1i1 binds Lhx3 and Isl1 separately or whether they form a complex of Lhx3-Tgfb1i1-Isl1. We found overexpressed Tgfb1i1 further enhanced the association between Lhx3 and Isl1 (*Figure 3C*). Lmo4, in contrast, induces a dose-dependent decrease in the association between Isl1 and Lhx3 (*Figure 3D*). Collectively, these results suggest Tgfb1i1 links Isl1 and Lhx3 to form a hetero-tetrameric (or larger) complex while Lmo4 interferes with the complex formation.

It is possible Pax6 interacts with the homeodomains of Isl1 and Lhx3 to form a Pax6-LIM protein complex, since Pax6 reportedly interacts with various homeodomain-containing proteins (*Granger et al., 2006*; *Mikkola et al., 2001*). We also found Pax6 interacts only with Isl1, but not Lhx3, via its paired domain (PD) (*Figure 3E*; *Figure 3—figure supplement 1G*). Both the HD and LIM domains of Isl1 participated to interact with Pax6, thus Pax6 might compete with Tgfb1i1 and Lhx3 to bind Isl1 (*Figure 3H*, top; *Figure 3—figure supplement 1D*).

The DF3 and DF4, which are separated by an auto-regulatory PBS, are respective targets of Isl1 and Lhx3 (*Figure 1A*; *Figure 1—figure supplement 3C,D*). Thus, Pax6 binding to the PBS may hinder the binding of Isl1-Tgfb1i1-Lhx3 complex to the DF3 and DF4 sequences, and vice versa. Using ChIP analyses in the cultured cells, we found Isl1, Lhx3, and Tgfb1i1 reduce the binding of Pax6 to the α-enhancer when all three are co-expressed but not when expressed individually (*Figure 3G*, three right graphs). Conversely, Pax6 expression interferes with the access of Tgfb1i1 to the α-enhancer (*Figure 3G*, rightmost graph). Pax6 does not affect Lhx3 binding to the α-enhancer, but it promotes Isl1 binding (*Figure 3H*, two center graphs). Together, these molecular interaction results suggest two different transcription factor complexes occupying the α-enhancer region. The Isl1-Pax6 complex binds to the DF3 and PBS and activates the α-enhancer (*Figure 3H*, top), whereas the Isl1-Tgfb1i1-Lhx3 complex binds to DF3 and DF4 to cover the area between those two sequences and inhibit the access of Pax6 to DF3 (*Figure 3H*, bottom).

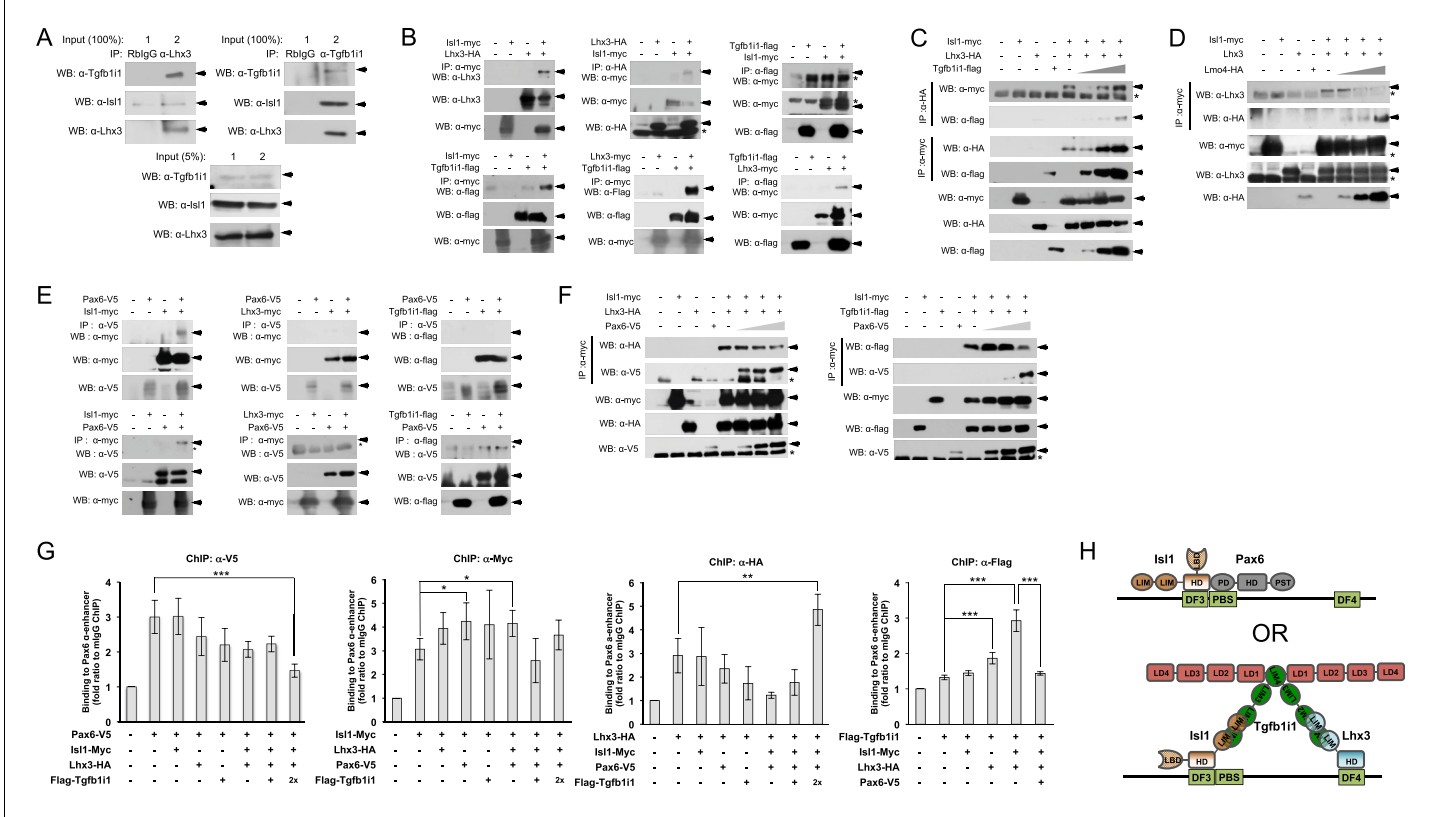

**Figure 3.** Pax6 and Tgfb1i1 antagonistically regulate Isl1-Lhx3 complex formation. (**A**) Interactions between endogenous Isl1, Lhx3, and Tgfb1i1 in P7 mouse retinas measured by reciprocal co-immunoprecipitation (co-IP) and subsequent Western blotting (WB) with the indicated antibodies. P7 mouse retinal cell lysates were divided into two input tubes (1 and 2) in prior to the co-IP with indicated antibodies and subsequent WB detection of co-immunoprecipitated proteins. 5% of input cell lysates were used to compare the relative amount of the proteins in the retinal cell lysates used for co-IP. (**B**) Interactions between epitope-tagged Lhx3 and Isl1, Lhx3 and Tgfb1i1, and Isl1 and Tgfb1i1 in HEK293T cells were determined by co-IP and WB. The successful expression of each transfected cDNA was determined by WB for each protein in cell lysates (50 μg/lane; 5% of the co-IP input) with the corresponding epitope-tag antibodies. Arrows indicate specific WB bands, and asterisks indicate non-specific bands. (**C** and **D**) The effects of Tgfb1i1 and Lmo4 on Isl1-Lhx3 complex formation in HEK293T cells. Triangles denote increasing amounts of each DNA construct (1 μg, 2 μg, and 4 μg). Interaction between Pax6 and LIM domain proteins (**E**) and effect of Pax6 on LIM domain protein complex formation (**F**) in HEK293T cells were also examined by co-IP and WB analyses. (**G**) Reciprocal effects of LIM domain proteins and Pax6 on the binding to human *PAX6* α-enhancer sequence in the transfected HEK293T cells were measured by qPCR amplification of α-enhancer sequences in DNA fragments isolated by ChIP with the indicated epitope tag-specific antibodies. Relative enrichment of each protein on the α-enhancer was determined by comparing the qPCR value of the transfected samples with those produced by antibodies bind non-specifically to the enhancer element in untransfected HEK293T cells. Error bars indicate STD (n > 5); *p<0.05, **p<0.01, ***p<0.001. (**H**) Schematic model depicting the binding of the Pax6-Isl1 and Isl1-Tgfb1i1-Lhx3 complexes to the *Pax6* α-enhancer element. HD, homeodomain; LBD, LIM-binding domain; LD, leucine-rich domain; PD, paired domain.

The following source data and figure supplement are available for figure 3:

**Source data 1.** Protein-protein interaction between LIM proteins.

**Figure supplement 1.** Pax6 and Tgfb1i1 antagonistically regulate Isl1-Lhx3 complex formation.

## Tgfb1i1⁻/⁻ retinas have excessive Pax6 α-enhancer-active GABAergic amacrine cells

We, next, examined the *Pax6* α-enhancer activity by detecting Pax6 α-GFP-positive cells in P14 *Tgfb1i1*⁻/⁻ mice in comparison to *Tgfb1i1*⁺/⁺ (wild-type, WT) littermates to validate the molecular mechanism proposed by our in vitro data in vivo. In support of the idea that Tgfb1i1 plays an important role in the inhibition of *Pax6* α-enhancer, the *Tgfb1i1*⁻/⁻ mouse retinas show more cells positive for the Pax6 α-GFP reporter than the retinas of their *Tgfb1i1*⁺/⁺ littermates (*Figure 4A,B*). We also

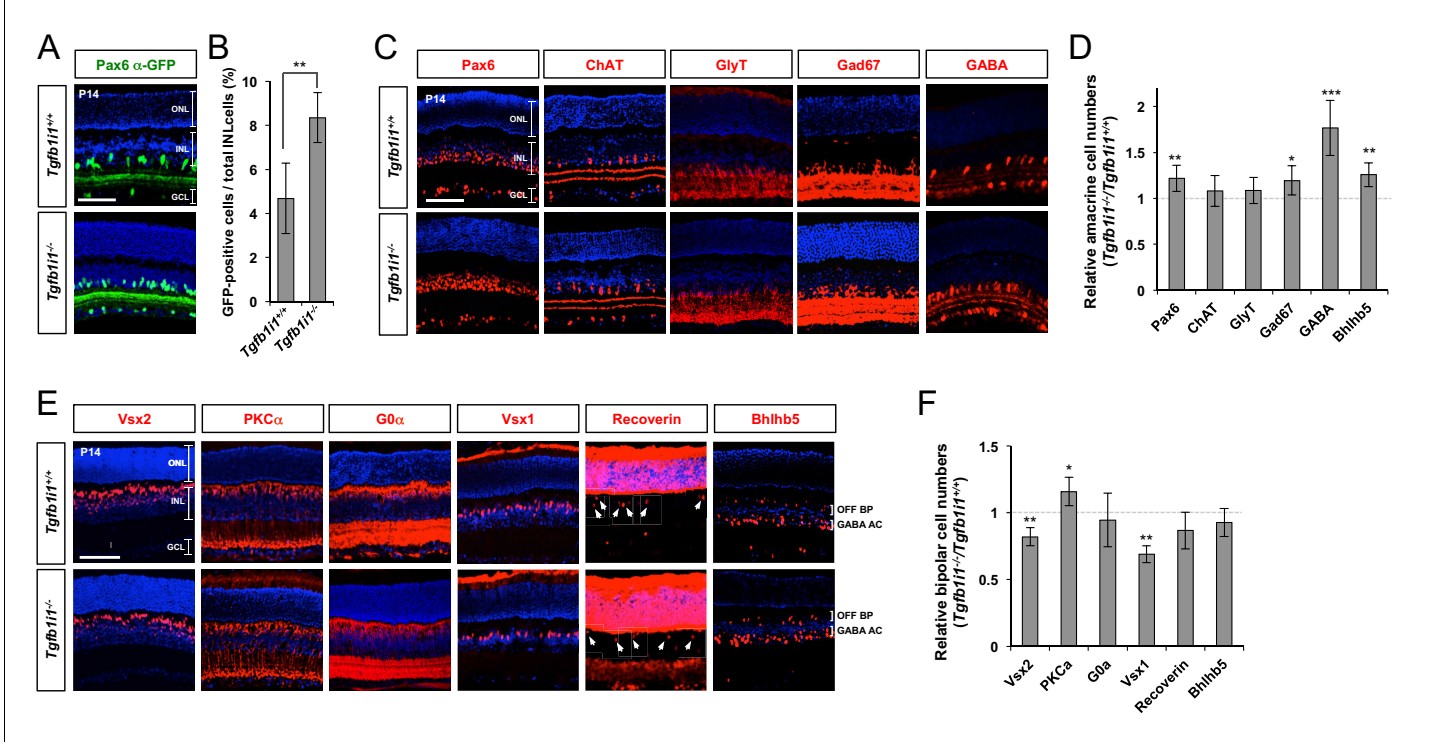

**Figure 4.** *Elevated GABAergic amacrine cell number in Tgfb1i1$^{-/-}$ mouse retinas.* (**A**) *Pax6* α-enhancer-active cells in P14 *Tgfb1i1$^{+/+}$* and *Tgfb1i1$^{-/-}$* littermate mouse retinas were visualized by immunodetection of GFP expressed from the *P6α-CreiGFP* transgene. ONL, outer nuclear layer; INL, inner nuclear layer; GCL, ganglion cell layer. (**B**) GFP-positive cell population in 250 μm x 250 μm retinal area. (**C**) P14 *Tgfb1i1$^{+/+}$* and *Tgfb1i1$^{-/-}$* littermate mouse retinas stained with antibodies detecting amacrine cell subtype-specific markers. Pax6, pan-amacrine cells; ChAT, cholinergic amacrine cells; GlyT1, glycinergic amacrine cells; GABA, Gad67 and Bhlhb5 (in the bottom half of INL in the images in **E**), GABAergic amacrine cells. (**D**) Fold-changes of amacrine cell numbers in P14 *Tgfb1i1$^{-/-}$* retinas compared to *Tgfb1i1$^{+/+}$* littermate retinas. (**E**) P14 *Tgfb1i1$^{+/+}$* and *Tgfb1i1$^{-/-}$* mouse retinas stained for bipolar cell-specific markers. Vsx2, pan-bipolar cell marker; PKCα, rod bipolar cells; G0α, rod and ON-cone bipolar cells; Vsx1, OFF bipolar cells; Recoverin, photoreceptors (in the ONL) and type-2 OFF bipolar cells (in the INL); Bhlhb5 (in the top half of INL), type-2 OFF bipolar cells. (**F**) Fold-changes in marker-positive cell numbers in *Tgfb1i1$^{-/-}$* retinas compared to *Tgfb1i1$^{+/+}$* littermate retinas. Values on the Y-axes of B, D, and F are averages. Error bars indicate STD (n = 4, three litters); *p<0.05; **p<0.01; ***p<0.001. Scale bars in the pictures, 100 μm.

The following figure supplements are available for figure 4:

**Figure supplement 1.** Elevation of *Pax6* α-enhancer-active GABAergic amacrine cells in *Tgfb1i1$^{-/-}$* mouse retinas.

**Figure supplement 2.** Deletion of *Lhx3* in the post-natal mouse retina.

examined the retinal composition of those littermate mice, and we only observed differences in the amacrine and bipolar cell populations among the major retinal cell types (*Figure 4C–F*; other retinal cell types are not shown). *Tgfb1i1$^{-/-}$* mouse retinas have more Pax6-positive amacrine cells and fewer Vsx2-positive bipolar cells than *Tgfb1i1$^{+/+}$* littermates (*Figure 4C–F*).

We classified amacrine cells positive for Gad67 (glutamate decarboxylase 67 kDa), GABA, and Bhlhb5 (basic helix-loop-helix domain containing, class B, 5) as GABAergic; cells positive for ChAT (choline acetyl transferase) as cholinergic; and cells positive for GlyT1 (glycine transporter 1) as glycinergic. Among those amacrine cell subtypes, only GABAergic amacrine cells showed a significant increase, while the numbers of cholinergic and glycinergic amacrine cells remain unchanged (*Figure 4C,D*). Moreover, the Pax6 α-GFP-positive cells are mainly GABAergic amacrine cells (*Figure 4C,D*; *Figure 4—figure supplement 1*), suggesting a positive relationship between *Pax6* α-enhancer activity and GABAergic amacrine cell fate and a negative role of Tgfb1i1 in this.

Among bipolar cell subtypes, the *Tgfb1i1$^{-/-}$* mouse retinas show fewer Vsx1-positive OFF bipolar cells without significant changes in G0α-positive ON bipolar cells, including PKCα-positive rod

bipolar cells (*Figure 4E,F*). However, the numbers of type-2 OFF bipolar cells, which are positive to Bhlhb5 and Recoverin, are not greatly different between *Tgfb1i1$^{+/+}$* and *Tgfb1i1$^{-/-}$* mouse retinas (*Figure 4C–F*; *Figure 4—figure supplement 1*). The results therefore suggest that Tgfb1i1 is necessary for the development of Vsx1-positive OFF bipolar cells, except for type-2 subset, in mouse retina.

We also tried to investigate the roles of Lhx3 in the post-natal mouse retina, which cannot develop in *Lhx3*-deficient mice that die perinatally (*Sheng et al., 1996*). We, thus, electroporated DNA constructs encoding Cas9 endonuclease and single guide RNA (sgRNA) targets to mouse *Lhx3* sequence, together with the pCAGIG DNA construct expressing EGFP (enhanced green fluorescent protein), into P0 mouse retinas (*Figure 4—figure supplement 2A*; see Materials and methods for details). We then examined the fates of EGFP-positive cells, which supposedly co-express Cas9 and the Lhx3 sgRNA, in the mouse retinas at P14. We found GABAergic amacrine cell identities of the retinal cells expressing the constructs were significantly enhanced, whereas OFF bipolar cell identities of the cells were remarkably diminished (*Figure 4—figure supplement 2B–D*). Collectively, our results suggest that Tgfb1i1 supports the development of OFF bipolar cell subsets, while it antagonizes the development of GABAergic amacrine cell subset, by forming Lhx3-containing protein complex that inhibits *Pax6* α-enhancer activity in post-natal mouse retina.

## Positive correlation between Pax6 α-enhancer-driven Pax6ΔPD expression and GABAergic amacrine cell fate

In P14 mouse retinas, Lhx3-positive bipolar cells co-expressing Isl1 comprise only 20% of total Lhx3-positive cells (*Figure 5—figure supplement 1B,C*). In contrast, 82% of Lhx3-positive retinal cells co-express Isl1 at P7, which is when Tgfb1i1 is expressed in most retinal cell types apart from amacrine cells (*Figure 5—figure supplement 1A,C*). This suggests the Isl1-Tgfb1i1-Lhx3 complex may form in the retinal cells around the first post-natal week at the peak of bipolar cell development (*Morrow et al., 2008*; *Rapaport et al., 2004*). Supporting this, the interaction between Isl1 and Lhx3 is significantly reduced in P7 *Tgfb1i1$^{-/-}$* retinas (*Figure 5A*, top). This might trigger an over-activation of *Pax6* transcription, driven by the α-enhancer.

However, Pax6 levels did not differ in *Tgfb1i1$^{-/-}$* and *Tgfb1i1$^{+/+}$* mouse retinas (*Figure 5B*, larger Pax6 bands), despite the increase of Pax6-positive cells in *Tgfb1i1$^{-/-}$* mouse retina (*Figure 4C,D*). We did, instead, observe a specific increase in the level of Pax6ΔPD isoform, which is an alternative transcript produced at downstream of the α-enhancer sequence (*Lakowski et al., 2007*; *Plaza et al., 1995*), in the *Tgfb1i1$^{-/-}$* mouse retina (*Figure 5B*, smaller Pax6 bands). This Pax6ΔPD isoform is selectively enriched in Pax6 α-GFP-positive cells purified from P7 *P6α-CreiGFP* retinas by fluorescence activated cell sorting (FACS) (*Figure 5C*). Our results, thus, suggest hyperactivation of the *Pax6* α-enhancer in *Tgfb1i1$^{-/-}$* retinas triggers ectopic expression of Pax6ΔPD isoform, but not the canonical Pax6.

We, next, investigated the role of α-enhancer-driven Pax6ΔPD expression in retinal cell fate determination by overexpressing Pax6ΔPD, which is connected with EGFP by internal ribosome entry site (IRES), in post-natal mouse retina (*Figure 5D–G*; *Figure 5—figure supplement 2*). About 54% of EGFP-positive INL cells in P14 mouse retinas, which were electroporated with the pCAGIG-Pax6ΔPD DNA at P0, are identified as Syntaxin-positive amacrine cells, whereas only 26% of EGFP-positive INL cells are amacrine cells in the retinas electroporated with control pCAGIG DNA (*Figure 5—figure supplement 2A* [top row], D). This is also contrary to the results of pCAGIG-Pax6-electroporated mouse retinas, in which about 85% of EGFP-positive INL cells are identified as amacrine cells (*Figure 5—figure supplement 2A* [top row, middle], D). Moreover, by showing insignificantly different marker positivity with EGFP;Syntaxin double-positive INL cells (54% ± 7.56% (Syntaxin) vs. 46% ± 7.33%(Gad67)), majority of EGFP-positive amacrine cells in the pCAGIG-Pax6ΔPD-electroporated retinas are predicted as GABAergic amacrine cells, which are approximately half of the EGFP;Syntaxin double-positive amacrine cell population in pCAGIG-Pax6-electroporated mouse retinas (85% ± 7.2%(Syntaxin) vs. 44% ± 9.17%(Gad67)) (*Figure 5F,G*; *Figure 5—figure supplement 2A* [second row], D). The populations of EGFP-positive cholinergic and glycinergic amacrine cells in pCAGIG-Pax6ΔPD-electroporated mouse retinas are not greatly different from those in pCAGIG-electroporated mouse retinas, but are lower than those in pCAGIG-Pax6-electroporated mouse retinas (*Figure 5—figure supplement 2A* [bottom two rows], D). Together, these results suggest that Pax6ΔPD preferentially supports GABAergic amacrine cell fate, while full-length

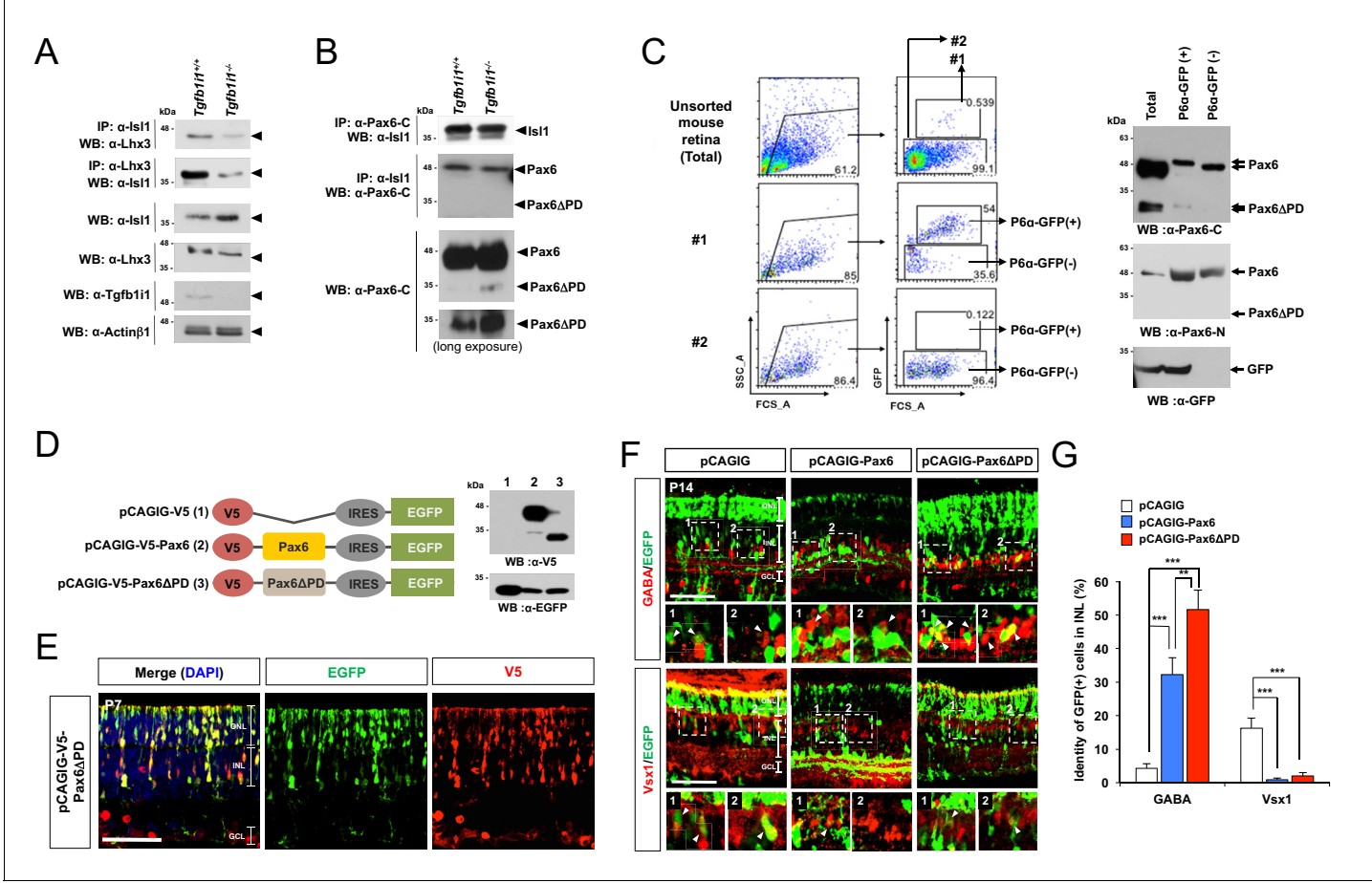

**Figure 5.** Pax6 α-enhancer-induced Pax6ΔPD isoform supports GABAergic amacrine cell fate. (**A**) Reciprocal co-IP and WB analyses with the indicated antibodies reveal a reduced interaction between Isl1 and Lhx3 in P7 $Tgfb1i1^{-/-}$ mouse retinas compared with littermate $Tgfb1i1^{+/+}$ retinas (top two WB images). $Tgfb1i1^{-/-}$ retinal lysates show 1.6-fold higher Isl1 level than $Tgfb1i1^{+/+}$ retinal lysates and no significant change in the levels of Lhx3 and Actinβ1 (bottom four WB images). (**B**) No significant difference in the assembly of Isl1 and Pax6 was observed in P7 $Tgfb1i1^{+/+}$ and $Tgfb1i1^{-/-}$ littermate mouse retinas (top two WB images). $Tgfb1i1^{-/-}$ retinas show higher expression of the Pax6ΔPD isoform than $Tgfb1i1^{+/+}$ retinas and no change in full-length Pax6 (bottom two WB images). (**C**) Pax6 α-enhancer-active cells were isolated from P14 $P6\alpha$-CreiGFP retinas by repeated FACS (see the Materials and Methods). Lysates of GFP(+) and GFP(-) retinal cells were then analyzed by SDS-PAGE and WB with a rabbit anti-Pax6 antibody. Successful purification of the cells was confirmed by WB detection of GFP in each fraction. (**D**) Diagram of pCAGIG DNA constructs encoding V5-tagged Pax6 (pCAGIG-V5-Pax6) and Pax6ΔPD (pCAGIG-V5-Pax6ΔPD). These constructs express EGFP from an IRES linked to the V5-Pax6 or V5-Pax6ΔPD cDNAs. This allowed for the confirmation of successful expression of the cDNAs in in P7 mouse retinas electroporated with the indicated pCAGIG DNA constructs at P0 by WB detection of EGFP and V5. (**E**) Co-expression of V5-Pax6ΔPD and EGFP in P7 mouse retinas was also determined by immunostaining with mouse anti-V5 (red) and chick anti-GFP (green) antibodies. (**F**) The identities of EGFP-positive retinal cells co-expressing Pax6 or Pax6ΔPD in P14 mouse retinas were determined by staining with antibodies against various amacrine and bipolar cell-specific proteins. The images are mouse retinal sections stained with anti-GABA (top) and anti-Vsx1 (bottom) antibodies. Arrowheads indicate cells positive to both of EGFP and the markers. Additional immunostaining results are provided in *Figure 5—figure supplement 2*. (**G**) EGFP-positive cells co-expressing each cell type-specific marker are shown as a percentage of total EGFP-positive INL cells. Values on the Y-axis are averages. Error bars indicate STD (n = 5); **p<0.01; ***p<0.001.

The following figure supplements are available for figure 5:

**Figure supplement 1.** Distribution of Isl1- and Lhx3-expressing cells in $Tgfb1i1^{+/+}$ and $Tgfb1i1^{-/-}$ mouse retinas.

**Figure supplement 2.** Ectopic expression of Pax6 isoforms in the post-natal mouse retinas.

Pax6 induces all amacrine cell types in a similar ratio observed in the normal mouse retina (*Voinescu et al., 2009*).

Mouse retinas expressing ectopic Pax6ΔPD show almost no EGFP-positive cells co-expressing OFF bipolar cell markers including Vsx1, Recoverin, and Bhlhb5 (*Figure 5F* [bottom row], G; *Figure 5—figure supplement 2B,D*). On the contrary, significant numbers of EGFP-positive cells co-expressed ON bipolar cell marker G0α in pCAGIG-Pax6ΔPD-electroporated mouse retinas, and the numbers are not significantly different from those in pCAGIG-electroporated samples (*Figure 5—figure supplement 2B,D*). EGFP-positive cells in pCAGIG-Pax6-electroporated mouse retinas, however, are almost absent of both ON and OFF bipolar cell marker co-expression (*Figure 5—figure supplement 2B,D*). The results therefore suggest that Pax6ΔPD inhibits only OFF bipolar cell fate, while full-length Pax6 suppress both ON and OFF bipolar cell fates.

## Pax6-dependent Pax6 α-enhancer activation is important for GABAergic amacrine cell fate maintenance

Next, to inactivate the α-enhancer, we generated *Pax6^{ΔPBS/ΔPBS}* mice by deleting the auto-stimulatory PBS in the α-enhancer using the CRISPR/Cas9 system (*Figure 6A*; see Materials and methods for details). Despite the *Pax6* α-enhancer being active in the mouse retina from embryo to adult, the gross morphologies of *Pax6^{ΔPBS/ΔPBS}* mouse eyes are indistinguishable from *Pax6^{+/+}* WT eyes (*Figure 6B*), implicating dispensable roles of *Pax6* α-enhancer-induced Pax6ΔPD expression in the eye and retinal development. However, in P14 *Pax6^{ΔPBS/ΔPBS}* retinas, the α-enhancer-driven GFP and Pax6ΔPD expression are reduced significantly, but not entirely abolished (*Figure 6B–D*). Since Pax6 does not bind and activate the *Pax6^{ΔPBS}* α-enhancer (*Figure 6—figure supplement 1*), this suggests the presence of positive regulator(s) of the α-enhancer in the mouse retina other than Pax6.

P14 *Pax6^{ΔPBS/ΔPBS}* retinas show significantly fewer GABAergic amacrine cells than the retinas of *Pax6^{+/+}* WT littermates, despite similar total numbers of Pax6-positive amacrine cells (*Figure 6E* [left two columns], F; *Figure 6—figure supplement 2*). Conversely, *Pax6^{ΔPBS/ΔPBS}* retinas show more OFF bipolar cells (i.e., Vsx1-positive), despite similar total numbers of Vsx2-positive bipolar cells (*Figure 6E* [right two columns], F; *Figure 6—figure supplement 2*). However, the numbers of GABAergic amacrine cells, which start to develop in the embryonic retina (*Voinescu et al., 2009*), were not significantly different between *Pax6^{+/+}* and *Pax6^{ΔPBS/ΔPBS}* retinas until P4 when the bipolar cells start to develop (*Figure 6—figure supplement 3A* [bottom three rows], B). The results therefore suggest that Pax6-dependent activation of *Pax6* α-enhancer is not essential for the embryonic development of GABAergic amacrine cells but it might be necessary for the development and/or maintenance of those cells in the post-natal retina.

To test a possibility of antagonistic fate determination of newborn retinal neurons between GABAergic amacrine and OFF bipolar cell subsets in the post-natal mouse retinas, we repeatedly injected bromodeoxyuridine (BrdU) to WT, *Tgfb1i1^{−/−}*, and *Pax6^{ΔPBS/ΔPBS}* mice to label GABAergic amacrine and OFF bipolar cells, which were born between post-natal day 4 and 7. We failed to find BrdU;Bhlhb5 double-positive GABAergic amacrine cells in P14 WT, *Tgfb1i1^{−/−}*, and *Pax6^{ΔPBS/ΔPBS}* mouse retinas, suggesting the lack of newborn GABAergic amacrine cells in the post-natal mouse retinas (*Figure 6—figure supplement 3C,E*). Furthermore, the number of BrdU;Vsx1 double-positive OFF bipolar cells in those mouse retinas were not significantly different each other (*Figure 6—figure supplement 3C* [bottom row], D), despite the remarkable decrease and increase of total Vsx1-positive cell numbers in P14 *Tgfb1i1^{−/−}* and *Pax6^{ΔPBS/ΔPBS}* in mouse retinas, respectively (*Figure 4E,F*; *Figure 6E,F*). The results therefore suggest that the alteration of OFF bipolar cells in those two mutant mouse retinas was not caused by neurogenic fate changes of newborn retinal cells but may have resulted from fate change of preexisting retinal cells.

We, thus, traced the fates of retinal cells born in the embryonic retina by injecting BrdU to pregnant female mice at 15 dpc (days post coitum). The numbers of Bhlhb5;BrdU-labeled GABAergic amacrine cells are significantly decreased in P7 *Pax6^{ΔPBS/ΔPBS}* mouse retinas in comparison to their WT littermate mouse retinas (*Figure 6—figure supplement 3D* [top row], F). Conversely, Vsx1; BrdU-labeled OFF bipolar cell numbers are significantly increased in the *Pax6^{ΔPBS/ΔPBS}* mouse retinas (*Figure 6—figure supplement 3D* [bottom row], F). Taken together, these results suggest Pax6 and the Isl1-Tgfb1i1-Lhx3 complex in the post-natal mouse retina competitively regulate the *Pax6* α-enhancer-driven expression of Pax6ΔPD, which maintains GABAergic amacrine cell fate against the transdifferentiation into OFF bipolar cells.

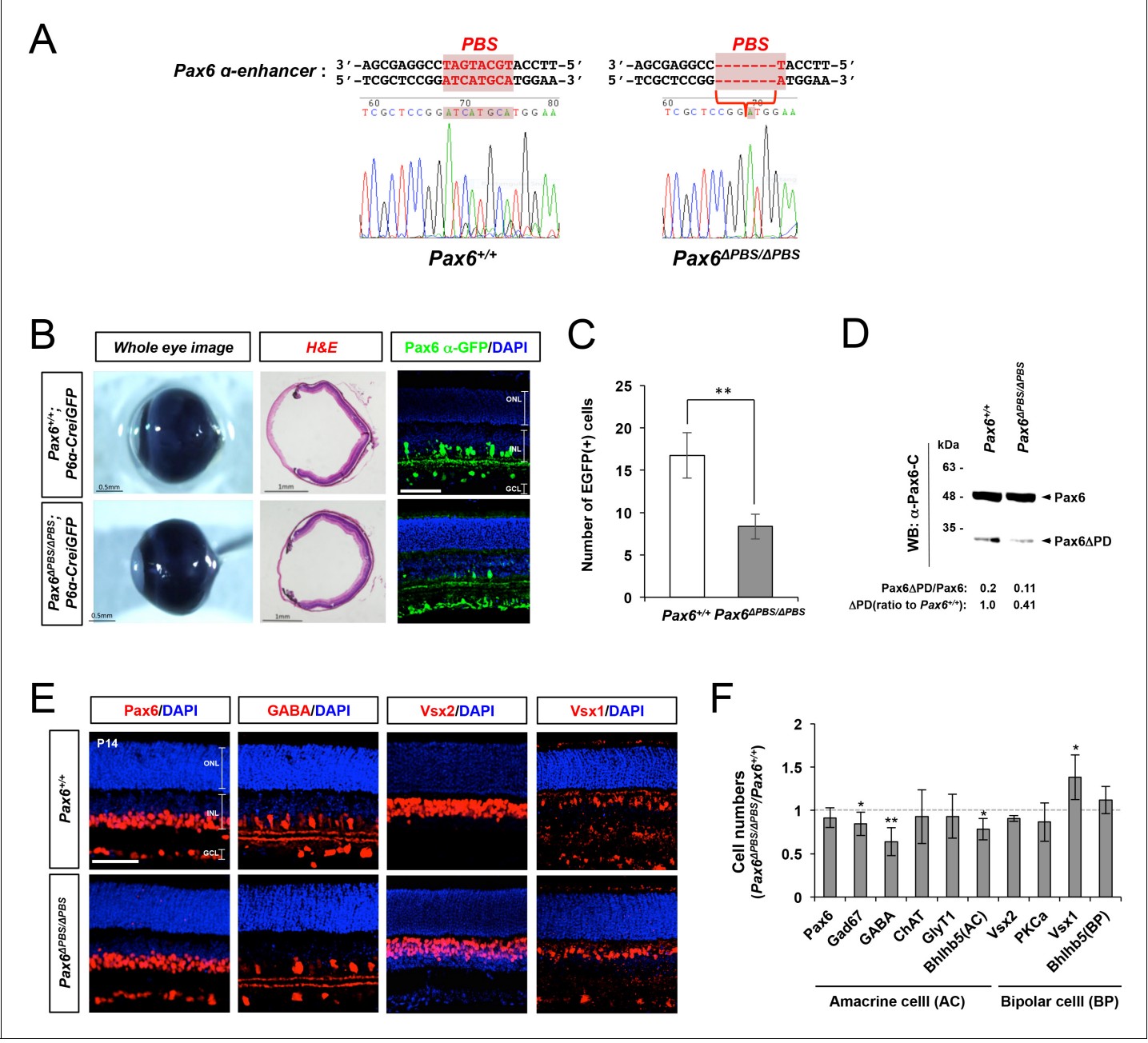

**Figure 6.** Pax6-dependent Pax6 α-enhancer activation is positively correlated with GABAergic amacrine cell number. (A) Genomic DNA was isolated from the tails of $Pax6^{+/+}$(left) and $Pax6^{\Delta PBS/\Delta PBS}$ (right) mice for sequencing the Pax6 α-enhancer region. The Pax6 binding sequence (PBS) in the α-enhancer is colored red. The $Pax6^{\Delta PBS}$ allele is missing six nucleotides (5′-TGCATG-3′) in the PBS. (B) Whole eye images of P30 $Pax6^{+/+}$;$P6\alpha$-CreiGFP and $Pax6^{\Delta PBS/\Delta PBS}$;$P6\alpha$-CreiGFP littermate mice (left) and the mouse eye sections stained with H&E (center) or an anti-GFP antibody (right). Scale bar in the rightmost column is 100 μm. (C) Pax6 α-GFP-positive cells in P30 $Pax6^{+/+}$and $Pax6^{\Delta PBS/\Delta PBS}$ retinas (250 μm x 250 μm). Error bars indicate STD (n = 4, two independent litters). (D) Full-length Pax6 and Pax6ΔPD in P14 $Pax6^{+/+}$ and $Pax6^{\Delta PBS/\Delta PBS}$ retinal cell lysates were detected by WB with anti-Pax6 antibody and WB band intensities were compared to show the relative values below the WB image. (E) Distributions of pan-amacrine cell marker Pax6, GABAergic amacrine cell subset marker GABA, pan-bipolar cell marker Vsx2, and OFF bipolar cell marker Vsx1 in P14 $Pax6^{+/+}$and $Pax6^{\Delta PBS/\Delta PBS}$ littermate retinas were visualized with immunostaining with antibodies recognizing respective markers. Scale bars, 100 μm. Additional images of amacrine and bipolar cell subtypes are shown in *Figure 6—figure supplement 2*. (F) Quantification of relative numbers of amacrine and bipolar cell subsets in mouse retinas. Error bars indicate STD (n = 5, three independent litters). *p<0.05; **p<0.01.

The following figure supplements are available for figure 6:

**Figure supplement 1.** Impaired response of the $Pax6^{\Delta PBS}$ α-enhancer to Pax6.

*Figure 6 continued on next page*

*Figure 6 continued*

**Figure supplement 2.** Distribution of amacrine and bipolar cell subsets in *Pax6⁺/⁺* and *Pax6^{ΔPBS/ΔPBS}* mouse retina.

**Figure supplement 3.** Fate determination of GABAergic amacrine cells and OFF bipolar cells in the post-natal mouse retinas.

## Visual adaptation of the retina is sensitive to Pax6 α-enhancer-active GABAergic amacrine cell number

We next determined whether these changes in the *Pax6* α-enhancer-active (P6α) GABAergic amacrine cell and OFF bipolar cell numbers influence visual responses in *Tgfb1i1⁻/⁻* and *Pax6^{ΔPBS/ΔPBS}* mice. Using the OptoMotry system (*Prusky et al., 2004*), we observed a significant reduction in visual acuity of P60 *Tgfb1i1⁻/⁻* mice compared to age-matched WT and *Pax6^{ΔPBS/ΔPBS}* mice (*Figure 7A*, graph). Upon the measurement of light response of a whole retina by electroretinogram (ERG), the amplitudes for the a- and b-waves in dark-adapted (scotopic) and light-adapted (photopic) ERG responses of P60 *Tgfb1i1⁻/⁻* and *Pax6^{ΔPBS/ΔPBS}* mouse eyes were, however, unaltered in comparison to those of WT littermate controls (*Figure 7—figure supplement 1*). The results suggest that the functions of photoreceptors (determined by ERG a-waves) and ON bipolar cells (determined by ERG b-waves) are intact in those mutant mice. In support of this, the numbers of photoreceptors and ON bipolar cells in P60 *Tgfb1i1⁻/⁻* and *Pax6^{ΔPBS/ΔPBS}* mouse retinas were not significantly different from those in their littermate WT mice (*Figure 7—figure supplement 2*). Therefore, the reduced visual acuity of *Tgfb1i1⁻/⁻* mice might be caused by either the changes of visual pathway components in the brain or the alterations of amacrine cells and RGCs at downstream of bipolar cells.

We, therefore, measured the light-evoked activity of individual retinal circuits by performing multi-electrode array (MEA) recordings of RGCs, which represent the final circuit component in the retina. We found an increase in the basal firing rate and mean spike number, but no change in maximum spike rate for the light-ON responses of P60 *Tgfb1i1⁻/⁻* retinas when compared to WT littermate controls (*Figure 7B* [top], C). Conversely, we observed a reduction in the basal firing rate, maximum spike rate, and mean spike number for the ON responses of P60 *Pax6^{ΔPBS/ΔPBS}* retinas (*Figure 7B* [bottom], D). Interestingly, a significant number of RGCs in P60 *Tgfb1i1⁻/⁻* retinas do not return to the resting state after a transient light response (*Figure 7B*, arrowhead). Considering the GABAergic identity of P6α amacrine cells and the increase of the cells in the *Tgfb1i1⁻/⁻* retinas (*Figure 4C,D*; *Figure 4—figure supplement 1*), it suggests this specific amacrine cell subset might disinhibit ON response by acting to other inhibitory retinal neurons in light-ON pathway (*Chávez et al., 2010*; *Demb and Singer, 2012*; *Eggers et al., 2013*).

The low visual acuity and sustained light response of *Tgfb1i1⁻/⁻* mouse retinas suggest that the hypersensitivity of the mice to light interferes with their detection of dark objects on brighter backgrounds (*Figure 7A*, predicted views). To test this hypothesis, we presented the mice with two different types of visual stimuli. First, we trained dark-adapted mice to associate a water reward with a flashing light stimulus. Then, we counted correct water-licking events in response to various intensities of flash light (*Figure 7—figure supplement 3A*). *Tgfb1i1⁻/⁻* mice not only learn this task faster than WT mice (*Figure 7—figure supplement 3B*), but they also show more sensitive detection of the light stimuli (*Figure 7E*). However, in a second visual task requiring mice to detect a drifting grate stimulus after a light stimulus, *Tgfb1i1⁻/⁻* mice perform worse than WT mice (*Figure 7F*; *Figure 7—figure supplement 3C,D*). This suggests *Tgfb1i1⁻/⁻* retinas are more slowly re-sensitized after light exposure than WT retinas. Conversely, the re-sensitization of *Pax6^{ΔPBS/ΔPBS}* retinas is significantly faster than WT retinas, despite being less sensitive to light (*Figure 7F,G*; *Figure 7—figure supplement 3*). These results are consistent with our MEA recordings, which showed sustained and transient light responses in *Tgfb1i1⁻/⁻* and *Pax6^{ΔPBS/ΔPBS}* RGCs, respectively (*Figure 7B–D*).

Collectively, the results suggest that P6α amacrine cells control the tone of light-ON retinal pathway. The overall tone of light-ON pathway was increased in *Tgfb1i1⁻/⁻* mouse retinas, which have extra P6α amacrine cells, whereas it is decreased in *Pax6^{ΔPBS/ΔPBS}* mouse retinas having reduced P6α amacrine cell number. Consequently, the light-ON pathway is augmented in *Tgfb1i1⁻/⁻* mouse retinas and attenuated in *Pax6^{ΔPBS/ΔPBS}* mouse retinas to make the retinas hypersensitive and

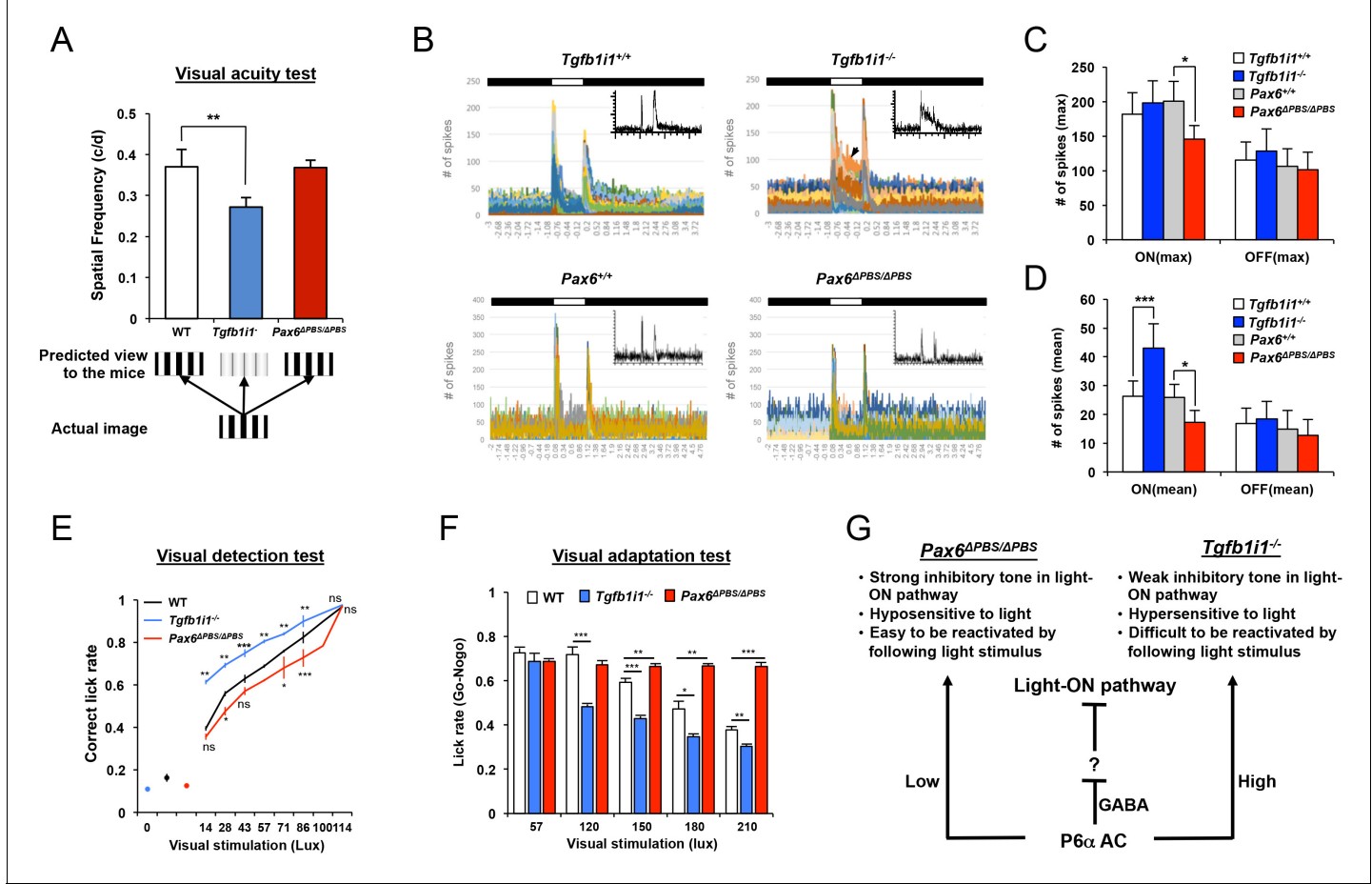

**Figure 7.** Pax6 α-enhancer-active amacrine cells are important for visual adaptation. (**A**) Visual acuity was measured in P60 mice using the OptoMotry system as previously described (*Prusky et al., 2004*) (for details, see the Materials and Methods). Error bars indicate STD (n = 6). \*\*p<0.01. (**B**) Peristimulus time histograms (PSTHs) for RGCs in P60 *Tgfb1i1+/+* and *Tgfb1i1−/−* littermate and P60 *Pax6+/+* and *Pax6ΔPBS/ΔPBS* littermate mouse retinas were obtained by multielectrode array (MEA) recordings. Maximum and mean numbers of spike were counted from each PSTH. Insets are representative PSTH patterns. Arrowhead indicates the sustained light-ON responses of RGCs. Maximum (max, **C**) and mean (**D**) numbers of spikes were counted from each PSTH. The numbers on the Y-axis are averages (WT, n = 526 (in four mice); *Tgfb1i1−/−*, n = 534 (in six mice); *Pax6+/+*, n = 175; *Pax6ΔPBS/ΔPBS*, n = 276). Error bars indicate STD. Statistical significance was determined using the D'Agostino and Pearson omnibus normality test followed by one-way ANOVAs and Sidak's test for multiple comparisons. \*p<0.05; \*\*\*p<0.001. (**E**) Visual detection in P60 *Tgfb1i1−/−*, *Pax6ΔPBS/ΔPBS*, and their WT littermate mice trained to lick water in response to light stimuli. The experimental scheme and task learning curves are provided in *Figure 7—figure supplement 3A and B* (for details, see the Materials and Methods). (**F**) The mice were also given water in association with a continuous light stimulus (2 s) but not with a continuous light stimulus (1 s) followed by a drifting grate stimulus (1 s) (see the experimental scheme and task learning curves in *Figure 7—figure supplement 3C and D*). Visual responses were quantified as ratios of hit rates (HitR, Go) to false alarm rates (FAR, Nogo). Error bars in (**E**) and (**F**) indicate STD. \*p<0.05; \*\*p<0.01; \*\*\*p<0.001 (Unpaired t-test). (**G**) Diagram depicting the modulation of retinal circuitry important for visual adaptation by *Pax6* α-enhancer-active (P6α) GABAergic amacrine cells.

The following figure supplements are available for figure 7:

**Figure supplement 1.** ERGs of mouse retinas.

**Figure supplement 2.** Cell composition of P60 WT, *Tgfb1i1−/−*, and *Pax6ΔPBS/ΔPBS* mouse retinas.

**Figure supplement 3.** Experimental scheme assessing mouse visual responses.

hyposensitive to light stimulus, respectively (*Figure 7B–E*). On the other hand, the *Tgfb1i1*$^{-/-}$ mouse retinas cannot be re-sensitized as fast as WT retinas, while the *Pax6*$^{\Delta PBS/\Delta PBS}$ mouse retinas can be re-sensitized more quickly than WT retinas after a light stimulus. Given the GABAergic identity of the cells, the P6α amacrine cells may inhibit the activity of post-synaptic partners, which are not identified yet but can be predicted as an inhibitory neuron in light-ON pathway (*Figure 7F*). Therefore, the results indicate that the proper number of P6α amacrine cells should be present in the retina to respond to light efficiently and adequately.

## Discussion

Transcription factors frequently act in combination, allowing relatively few to generate the tremendous cellular diversity of the nervous system (*Jessell, 2000*). Especially, the 'LIM code' mixes and matches LIM domain-containing transcription factors to direct tissue- and cell-specific gene expression (*Gill, 2003*; *Shirasaki and Pfaff, 2002*). Lhx3, for example, specifies motor neuron cell fate in the spinal cord by forming a hetero-hexameric complex with Isl1 and nuclear LIM interactor (NLI) for the binding to the promoter of the *Mnx1/Hb9* gene, whereas it specifies V2 interneuron cell fate by forming a hetero-tetrameric complex with NLI at the promoter of the *Vsx2/Chx10* gene (*Thaler et al., 2002*). Given that the various LIM homeodomain transcription factors, including Lhx2, Lhx3, Lhx4, and Lhx9, share a consensus target sequence (*Gehring et al., 1994*), we speculate Isl1 partners with different LIM homeodomain transcription factors in a cell-context-dependent manner. In contrast to its relationship with Lhx3, Isl1 cooperates with Lhx2 to activate the α-enhancer in cultured cell lines (data not shown). However, this is unlikely to occur in vivo, because Lhx2 and Isl1 are expressed mutually exclusively in RPCs (Lhx2) and post-mitotic RGCs (Isl1) of the embryonic mouse retina, in GABAergic (Lhx2) and cholinergic amacrine cells (Isl1) in the mature retina, as well as in Müller glia (Lhx2) and ON bipolar cells (Isl1) (*Balasubramanian et al., 2014*; *Elshatory et al., 2007*; *Gordon et al., 2013*; *Pan et al., 2008*). Moreover, *Lhx2*$^{flox/flox}$;*P6α-Cre* retinas, which lack Lhx2 expression in the Cre-active lineages (*Gordon et al., 2013*), show no change in the number of *Pax6* α-enhancer-active cells (data not shown). This suggests Lhx2 may be dispensable for the activation of the *Pax6* α-enhancer in the mouse retina.

We propose that a Tgfb1i1 dimer links Isl1 and Lhx3 to form a hetero-tetrameric complex that represses the *Pax6* α-enhancer (*Figures 2* and *3*). The effects of Tgfb1i1 on the α-enhancer could be achieved by blocking Pax6's access the PBS sequence (*Figure 3G,H*). Alternatively, Tgfb1i1 may also recruit transcriptional co-repressors, such as NCoR (nuclear receptor co-repressor) (*Heitzer and DeFranco, 2006*), to the *Pax6* α-enhancer. These negative effects of Tgfb1i1 on the *Pax6* α-enhancer can be antagonized by Lmo4, which is persistently co-expressed with Pax6 in the retina and interferes with the interactions between Tgfb1i1 and Lhx3 and/or Isl1 (*Duquette et al., 2010*) (*Figures 2D* and *3D*). Retinas lacking *Lmo4* have fewer GABAergic amacrine cells than controls (*Duquette et al., 2010*), which suggests Lmo4 may positively affect *Pax6* α-enhancer-dependent GABAergic amacrine cell fate determination by inhibiting the formation of the LIM complex. However, the antagonistic regulation of the LIM complex by Tgfb1i1 and Lmo4 could not be applied to OFF bipolar cell fate determination, since OFF bipolar cell numbers are decreased commonly in *Tgfb1i1*$^{-/-}$ and Lmo4-cko mouse retinas. Our results suggest that Tgfb1i1 and Lmo4 might involve in the development of different OFF bipolar cell subsets. The numbers of Bhlhb5-positive OFF bipolar cell subsets were not altered significantly in *Tgfb1i1*$^{-/-}$ mouse retinas (*Figure 4E,F*), in contrast to a significant decrease in Lmo4-cko mouse retinas.

In addition to its canonical form, two alternative forms of Pax6, Pax6(5a) and Pax6ΔPD, are produced by alternative splicing and internal transcription initiation, respectively (*Epstein et al., 1994*; *Mishra et al., 2002*). Pax6ΔPD does not affect Pax6 target gene expression via the conserved PBS (data not shown). Instead, as previously reported (*Mikkola et al., 2001*), Pax6ΔPD may potentiate the expression of Pax6 target genes by interacting with full-length Pax6. This facilitation of Pax6-induced gene transcription by Pax6ΔPD may also occur with the *Pax6* α-enhancer, resulting in a feed-forward activation of the α-enhancer. Alternatively, it may bind another promoter element containing the Pax6 homeodomain target DNA sequence (TAATT(/C)NA(/C)ATTA). Therefore, future studies will be needed to identify the targets of Pax6ΔPD in RPCs and post-mitotic retinal neurons. This will provide a full understanding of the distinctive roles Pax6 and Pax6ΔPD play in the retina.

Although the mechanisms of light adaptation and re-sensitization in the photoreceptors are fairly well-understood, how the inner retina contributes to these mechanisms is less clear. Acting downstream of rod bipolar cells that deliver visual signals from rod photoreceptors, A17 GABAergic amacrine cells provide a direct feedback inhibition to the rod bipolar cells (*Chávez et al., 2010*). In parallel, an unidentified subset of GABAergic amacrine cells is also proposed to inhibit rod bipolar cells at the downstream of ON-cone bipolar cells, which can be activated by AII amacrine cells in the rod pathway as well as by daylight (*Demb and Singer, 2012*; *Eggers et al., 2013*). GABAergic inhibition to the rod bipolar cells could be reduced in $Tgfb1i1^{-/-}$ mouse retinas, leading to sustained ON responses (*Figure 7A*). Conversely, the ON pathway in $Pax6^{\Delta PBS/\Delta PBS}$ mouse retinas is activated more transiently and is also more readily re-activated by subsequent visual stimuli (*Figure 7D*). Therefore, the P6α amacrine cells might attenuate those GABAergic inhibitions to rod bipolar cells and prevent premature inactivation of rod pathway. However, future studies should identify molecular and electrophysiological identities of the P6α amacrine cells and their pre- and post-synaptic partners to fully understand this visual adaptive circuits in the inner retina.

## Materials and methods

### DNA constructs and mouse lines

cDNA clones were generous gifts from Dr. Motoko Shibanuma (Hic-5/Tgfb1i1) and Dr. Seth Blackshaw (Lhx2 and Lhx9). The full-length and fragment DNAs used in this study were isolated by PCR amplification from these cDNAs. $Tgfb1i1^{-/-}$ mice were generated as previously described (*Kim-Kaneyama et al., 2011*). $Pax6^{\Delta PBS/\Delta PBS}$ mice were generated in this study using the CRISPR/Cas9 system (*Cong et al., 2013*). To prepare sgRNA constructs, the pX330 vector was obtained from Addgene and digested with BbsI for insertion of a pair of phosphorylated dsDNA oligos (5'-CACC-GAAGTCGCTCCGGATCATGCA-3', 5-AAACTGCATGATCCGGAGCGACTTC-3') that target the *PBS* in the *Pax6* α-enhancer. A T7 promoter was added to the 5' end of the sgRNA sequence in the pX330-sgRNA construct. This construct was then used as a template for in vitro transcription using the MEGAshortscript T7 kit (Life Technologies, CA). The in vitro transcribed sgRNAs (50 ng/ml) were injected into C57BL6/J mouse embryos (2 cell-stage) together with Cas9 mRNA (100 ng/μl; purchased from Toolgen Inc., South Korea). Then, these embryos were injected into the inner cell mass of ICR embryos. Four resulting F1 chimeric male mice were crossed to C57BL6/J female mice to obtain an F2 generation with the potential to carry deletions in the PBS. Then, tail DNA from each F2 mouse (n = 51) was prepared and used as a template for the PCR-amplification of the *Pax6* α-enhancer sequence. Each resulting PCR product was cloned into the pGEM-T vector for sequencing. Four F2 mice carry different heterozygous deletions in the PBS sequence were obtained. Before breeding with littermate $Pax6^{+/\Delta PBS}$ female mice, $Pax6^{+/\Delta PBS}$ male mice were crossed with C57BL6/J females for more than six generations to dilute any potentially OFF target mutations. All experiments using mice were performed according to the regulations of the KAIST-IACUC (KA2012-38).

### Cell culture and luciferase assay

HEK293T (RRID: CVCL_0063) and R28 retinal progenitor cells (RRID: CVCL_5I35) were obtained from ATCC and a gift from Dr. Gail Seigel (University of Rochester School of Medicine and Dentistry), respectively. These cell-lines are not in the list of commonly misidentified cell lines (by the International Cell Line Authentication Committee). These cells were regularly checked for mycoplasma contamination. The cells were maintained in DMEM supplemented with 10% fetal bovine serum (GIBCO, MA). Cells were combinatorially transfected with DNA constructs via the PEI (polyethylenimine) method (Polyscience, PA). The PCR-amplified mouse *Pax6* α-enhancer sequence was fused to the pGL3-luciferase vector (Promega, WI) and co-transfected with DNA constructs of interest and pSV-*β*-gal plasmids. Transfected cells were harvested at 24 hr after transfection, and cell extracts were assessed for luciferase activity followed by normalization using *β*-galactosidase activity.

### DNA affinity-capture assay

The $(CA)_5$ or $(TG)_5$ ssDNA oligonucleotides were coupled to CNBr-preactivated Sepharose 4B (GE Healthcare, IL) according to the manufacturer's protocol. R28 cells ($\sim 10^8$) were incubated in a low salt buffer (10 mM HEPES, 10 mM KCl, 0.1 mM EDTA, 1 mM DTT, and 0.5 mM PMSF) on ice for 10

min to rupture the plasma membranes. Then, the nuclei were collected by centrifugation. After treating the isolated nuclei with 10% (final v/v) NP-40 for 20 min, a solution containing 20 mM HEPES, 0.4 M NaCl, 1 mM EDTA, 1 mM DTT, and 1 mM PMSF was added to extract nuclear protein complexes. These nuclear extracts were then pre-cleared with a 5% (final v/v) slurry of protein A agarose beads (Invitrogen, CA) in a binding buffer (10 mM Tris-Cl pH 7.5, 0.4 M NaCl, 1 mM EDTA, 1 mM DTT, and 5% glycerol) at 4°C for 30 min.

The DF4 dsDNA oligonucleotides with 5'-(GT)$_5$ single-strand overhangs were synthesized with a 5' terminal amine modification and incubated with the R28 nuclear extracts overnight at 4°C with agitation. The protein-DF4 dsDNA complexes were then incubated with the ssDNA-coupled Sepharose 4B for 6 hr at 4°C, centrifuged, washed twice in binding buffer, washed twice in wash buffer (10 mM Tris-Cl pH 7.5, 1.2 mM NaCl, 1 mM EDTA, 1 mM DTT, 0.1% Nonidet P-40), and washed twice in PBS. Protein-DF4 complexes bound to the column were then eluted in SDS sample buffer for SDS-PAGE and subsequent silver staining. The silver-stained protein bands, which were enriched in the (CA)$_5$-coupled Sepharose 4B relative to the (TG)$_5$-coupled Sepharose 4B, were isolated for trypsin digestion before being subjected to MALDI-TOF MS/MS analysis at the Korean Basic Science Institute (KBSI) proteomics core facility.

## Electrophoretic mobility shift assay (EMSA)

Biotin-labeled and unlabeled dsDNA probes in binding buffer (75 mM NaCl, 1 mM EDTA, 1 mM DTT, 10 mM Tris-HCl (pH 7.5), 6% glycerol, 2 mg BSA, and 500 ng poly (dI-dC)) were incubated on ice for 30 min with LIM proteins produced using the TNT Quick Coupled Transcription/Translation kit (Promega, WI). The EMSA was carried out on a 6% polyacrylamide gel in 0.5X TBE buffer. The DNA-protein complexes were then transferred to a nylon membrane (GE Healthcare, IL), and the biotin-labeled probes were detected using the Phototop-Star Detection Kit (New England BioLabs, MA) according to the manufacturer's recommendations.

## Co-immunoprecipitation

P7 mouse retinas and transfected HEK293T cells were lysed in a buffer consisting of 10 mM Tris-HCl (pH 7.4), 200 mM NaCl, 1% Triton X-100, 1% NP-40, and a protease inhibitor cocktail (Invitrogen, CA). Cell lysates were centrifuged at 12,000 g for 10 min at 4°C. The resulting supernatants were incubated with appropriate antibodies at 4°C for 16 hr, and then pre-washed protein A/G-sepharose (GE Healthcare, IL) was added to the samples. The protein A/G-sepharose immune complexes were washed five times with cell lysis buffer and subjected to SDS-PAGE and Western blotting (WB) for detection of co-immunoprecipitated proteins.

## Chromatin immunoprecipitation (ChIP)

P7 mouse retinas were isolated, chopped, and cross-linked with 1% formaldehyde in PBS for 10 min at room temperature. After a 5 min incubation in 125 mM glycine, the tissues were homogenized and the nuclei were isolated. These nuclei were then subjected to sonication to break their chromatin into ~600 bp fragments in a lysis buffer containing 50 mM Tris-HCl (pH 7.5), 150 mM NaCl, 5 mM EDTA, 0.5% NP-40, 1% Triton X-100, and a protease inhibitor cocktail (Invitrogen, CA). After pre-clearing with protein A agarose beads for 1 hr, the nuclear extracts were incubated for 16 hr with 1 μg of the appropriate antibody followed by incubation with protein A beads for 45 min at room temperature. The immune complexes were then washed three times with lysis buffer and then three more times with the same wash buffer containing 500 mM LiCl. After adding a Chelex 100 slurry to the washed beads, the DNA fragments were eluted for use as templates for qPCR. We used specific primers to amplify sequences in the ectoderm enhancer (fp1, 5'-CTAAAGTAGACACAGCCTT; rp1, 5'-GGAGACATTAGCTGAATTC) and the α-enhancer (fp2, 5'-GTGACAAGGCTGCCACAAGCGCC, rp2, 5'- CCGTGTCTAGACAGAAGCCCTCTC) of the mouse *Pax6* gene. qPCR was performed using the iTaq fast SYBR Green Master Mix (BioRad, CA) with these same primers and analyzed using the CFX-Manager software (Bio-Rad, CA). Gene expression was normalized to that of a sample containing only protein A beads.

## Immunohistochemistry

Frozen sections (12 μm) of embryonic heads and post-natal mouse eyes were incubated for 1 hr in a blocking solution containing 5% normal donkey serum and 5% normal goat serum in PBS containing 0.2% Triton X-100. The sections were incubated with the antibodies listed in *Table 1* for 16 hr at 4°C. Fluorescent images were obtained with a confocal microscope (Olympus FV100 and Zeiss LSM710) after staining with Cy3, Alexa 647, and Alexa 488-conjugated secondary antibodies at room temperature for 1 hr.

## Fluorescence-activated cell sorting (FACS)

*P6α-CreiGFP* adult mouse eyes were dissected and placed in Hank's Balanced Salt Solution (HBSS; Life technologies) to remove the lens. Retinas were peeled from the eyes and placed in 1 ml HBSS containing activated 10 mg/ml papain (Sigma-Aldrich) for 5 min at 37°C. Retinal cells were resuspended in HBSS with 2% FCS followed by centrifugation at 1600 rpm for 2 min. Cell pellets were then gently triturated in HBSS with 2% FCS, filtered through a 70 μm Filcons membrane prior to

**Table 1.** Antibody used in this study.

| Antigen | Species | Producer | Dilution |
|---|---|---|---|
| Bhlhb5 | Goat | Santa Cruz | 1:100 |
| Brn3b | Goat | Santa Cruz | 1:200 |
| Calbindin | Mouse | Sigma | 1:200 |
| Calretinin | Mouse | Millipore | 1:1000 |
| ChAT | Goat | Millipore | 1:200 |
| Isl1 | Rabbit | gift from Dr. Mi-Ryoung Song | 1:500 |
| Isl1 | Guinea Pig | gift from Dr. Mi-Ryoung Song | 1:10,000 |
| Gad67 | Mouse | Millipore | 1:500 |
| GABA | Guinea Pig | Millipore | 1:300 |
| GFAP | Rabbit | Abcam | 1:500 |
| GFP | Chick | Abcam | 1:200 |
| GFP | Rabbit | Santa Cruz | 1:500 |
| GlyT1 | Rabbit | Abcam | 1:200 |
| G0α | Mouse | Millipore | 1:300 |
| G/R opsin | Rabbit | Millipore | 1:200 |
| Lhx2 | Goat | Santa Cruz | 1:200 |
| Lhx3 | Rabbit | Abcam | 1:1000 |
| Lhx9 | Rabbit | Santa Cruz | 1:500 |
| Pax6 | Rabbit | Abcam | 1:200 |
| Pax6 | Rabbit | Covance | 1:300 |
| PKCα | Mouse | Sigma | 1:200 |
| Recoverin | Rabbit | Chemicon | 1:200 |
| Rhodopsin | Mouse | Millipore | 1:500 |
| Sox2 | Goat | Santa Cruz | 1:100 |
| Sox9 | Rabbit | Santa Cruz | 1:200 |
| Tgfb1i1(Hic-5) | Mouse | BD | 1:100 |
| Tgfb1i1(Hic-5) | Rabbit | Abcam | 1:100 |
| Vsx1 | Goat | Santa Cruz | 1:50 |
| Vsx2(Chx10) | Mouse | Santa Cruz | 1:200 |
| V5 | Mouse | Genway Biotech | 1:1000 |

FACS analysis. GFP-positive retinal cells were then sorted in an Aria Fusion Cell Sorter (Becton Dickinson) at 495 nm excitation and 519 nm emission. Following FACS analysis, cells were collected by centrifugation at 1600 rpm for 5 min and the cells were lysed in a buffer containing 10 mM Tris-HCl (pH 8.0), 1 mM EDTA, 1% Triton X-100, 0.1% SDS, and 150 mM NaCl.

## Subretinal DNA electroporation

Electroporation experiments were performed as previously described (*Matsuda and Cepko, 2004*). Approximately 0.5 µl (total; 5 µg/µl) DNA solution mixed with fast green dye was injected into the subretinal space of P0 mouse retinas, and square electric pulses were applied (100 V; five 50 ms pulses at 950 ms intervals). For CRISPR/Cas9-mediated deletion of *Lhx3* gene, dsDNA oligos (sgRNA-Lhx3-1, 5'-(P)-CACCGGACCCGTCCCGGGAATCCGC-3' and 5'-AAACGCGGA TTCCCGGGACGGGTCC-3'; sgRNA-Lhx3-2, 5'-CACCGTGCTGGCGTTGTTGGCGCGA-3' and 5'-AAACTCGCGCCAACAACGCCAGCAC-3') were cloned into the pX330 vector before co-electroporation with the pCAGIG vector (molar ratio of pX330 constructs to pCAGIG is 1:0.5).

## Multielectrode array (MEA) recordings

Mouse retinas were cut into 3 mm x 3 mm patches in artificial cerebrospinal fluid (ACSF) solution (124 mM NaCl, 10 mM glucose, 1.15 mM $KH_2PO_4$, 25 mM $NaHCO_3$, 1.15 mM $MgSO_4$, 2.5 mM $CaCl_2$, and 5 mM KCl) bubbled with 95% $O_2$ + 5% $CO_2$ at pH 7.3–7.4°C and 32°C. Retinal patches were then mounted, ganglion cell layer down, on a planar $8 \times 8$ MEA, and the light-evoked RGC spikes were recorded using the MEA60 system (Multi Channel Systems GmbH, Germany). White light stimuli were applied with a DLP projector (Hewlett Packard, ep-7122) focused onto the photoreceptor layer of the retina through four convex lenses. Light intensity was 170–200 µW/cm$^2$ (116–136 lux) in 8–10 µW/cm$^2$ (5.5–6.8 lux) background illumination. Light stimuli were given in 1 s pulses with 6 s inter-pulse intervals to a total of 40 pulses per retina. All experiments were performed after sufficient dark adaptation (>1 hr).

## Visual acuity test

Mouse visual acuity was measured with the OptoMotry system (Cerebral Mechanics Inc.) as previously described (*Prusky et al., 2004*). Mice, of which genotypes are not determined before the measurement, were adapted to ambient light for 30 min and then placed on the stimulus platform, which is surrounded by four computer monitors displaying grating patterns randomly presented by the OptoMotry software. Mice that stopped moving and began tracking the gratings with reflexive head movements in concert with their rotation were counted as successful visual detection events. The detection thresholds were then obtained from the OptoMotry software.

## Visual performance test

### A.surgery

Adult mice (postnatal days 35–40) were anesthetized with isoflurane (1.5% induction and 1.0% maintenance) and fixed to a stereotaxic frame. Body temperature was maintained at 37°C. Custom-designed head plates were attached to the skull with small screws (Small Parts) and dental cement (Lang Dental).

### B.behavior test

Visual detection task

In this task, head plate-implanted mice (P45-P80) were trained to lick a water nozzle when they detected a visual stimulus (114 Lux). All mice used for this task were water-deprived for 1 day before beginning the training protocol. For the visual stimulus, we presented a full-field flashing light five times (10 Hz for 500 ms) through a gamma-corrected LCD monitor placed 10 cm from the left eye. Each stimulus trial began with a visual stimulus (500 ms in duration) and ended with a 2 s inter-trial period. Non-stimulus trials were randomly interleaved with stimulus trials using custom code (Presentation). We detected each lick through a custom-made lickport (4.0 mm ID) using a transistor-based lickometer system. Licks detected during the final 2 s of stimulus trials (i.e., in the response window) were rewarded with 4 µl of water (on). We delivered water rewards by gravity into the lickport under the control of a solenoid valve. Licks detected during the response window of non-stimulus trials

were punished with a mild air puff (300 ms) and a longer inter-trial interval (8 s, timeout) (off). We delivered compressed air puff punishments (10 psi) through a plastic tip (1.0 mm ID) located 2 cm from the face and controlled by a solenoid valve. Mice whose spontaneous lick rate during non-stimulus trials fell below 0.4 were advanced to the next phase. In the next phase, we measured lick rates in response to nine different intensities of visual stimuli presented randomly with equal probability.

### Visual adaptation task

We trained mice to discriminate a continuous light from a drifting grate stimulus presented after continuous light under a simple go/no-go protocol. Training proceeded in two steps: conditioning and discrimination. For conditioning (2–8 days), we trained each mouse to lick in response to continuous light. Each trial began with a continuous light (go stimulus, 2 s duration) and ended with an inter-trial period of 2 s. Licks detected in the final 2 s of the trial (i.e., in the response window) were rewarded with 10 µl of water for 2 s. During the conditioning phase, water rewards were still given after continuous light even if the animal failed to lick during the response window. Mice exceeding 300 licks within 1 hr were advanced to the discrimination phase. For discrimination (15 days), we trained mice to lick only when continuous light (2 s) was presented (go trial) and not to lick when a drifting horizontal grate stimulus (1 s) was presented after a continuous light stimulus (1 s) (no-go trial). All visual stimuli used for training were fixed at 57 Lux. We never presented the same type of stimulus more than three consecutive times. Licks within the response window of go trials were rewarded with water (4 µl) for 2 s, and licks within the response window of no-go trials were punished with mild air puffs (300 ms) and a longer inter-trial interval (8 s, timeout). Mice were neither rewarded nor punished for misses (i.e., no lick in a go trial) or correct rejections (i.e., no lick in a no-go trial). Training ended when the mouse stopped licking for 10 consecutive go trials.

Mice that reached threshold performance (lick rates in no-go trials < 0.4) were advanced to the next phase. In the next phase, we presented five different intensities of a continuous light higher or equal to 57 lux before a drifting grate stimulus fixed at 57 lux. We presented all stimuli randomly with equal probability.

## Acknowledgements

We thank Drs. Samuel Pfaff, Motoko Shibanuma, and Seth Blackshaw generous gifts of the research materials used in the study. We also thank Dr. Jong Soon Choi for the help with MALDI-TOF mass spectrometry and Dr. Suk-Jo Kang and Jiyoun Min for their help with FACS. This work was supported by grants from the Global Research Laboratory Program (NRF-2009–00424; JWK), Brain Research Program (NRF-2013–056566; JWK), Basic Science Research Program (NRF-2014R1A2A2A01003069; JWK), and Stem Cell Research Program (NRF-2006–2004289; KHK) funded by the Korean National Research Foundation (NRF); and by National Institute of Health of United States (NIH R01-EY013760; EML).

## Additional information

### Funding

| Funder | Grant reference number | Author |
| --- | --- | --- |
| National Research Foundation of Korea | NRF-2009-00424 | Jin Woo Kim |
| National Research Foundation | NRF-2006-2004289 | Kyung Hwa Kang |
| National Eye Institute | NIH R01-EY013760 | Edward M Levine |
| National Research Foundation of Korea | NRF-2013-056566 | Jin Woo Kim |
| National Research Foundation of Korea | NRF-2014R1A2A2A01003069 | Jin Woo Kim |

The funders had no role in study design, data collection and interpretation, or the decision to submit the work for publication.

## Author contributions

YK, Data curation, Software, Formal analysis, Validation, Investigation, Visualization, Methodology, Writing—original draft, Writing—review and editing; SL, TH, Data curation, Validation, Investigation, Visualization; Y-HS, Data curation, Validation, Investigation, Methodology; Y-IS, S-SP, Data curation, Formal analysis, Validation, Investigation; D-JP, AL, Data curation, Validation, Investigation; J-rK-K, Resources, Writing—original draft; M-RS, Resources, Writing—review and editing; EML, Resources, Data curation, Supervision, Writing—review and editing; I-BK, Data curation, Supervision, Methodology; YSG, Supervision, Methodology; S-HL, Data curation, Supervision, Methodology, Writing—original draft; KHK, Data curation, Supervision, Validation, Investigation; JWK, Conceptualization, Formal analysis, Supervision, Funding acquisition, Writing—original draft, Project administration, Writing—review and editing

## Author ORCIDs

Seung-Hee Lee, http://orcid.org/0000-0002-9486-5771
Jin Woo Kim, http://orcid.org/0000-0003-0767-1918

## Ethics

Animal experimentation: All experiments using mice were performed according to the regulations of the KAIST-IACUC (KA2012-38).

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
