## [Decision Letter]

Thank you for submitting your article "The LIM protein complex establishes a retinal circuitry of visual adaptation by regulating Pax6 α-enhancer activity" for consideration by *eLife*. The manuscript has been reviewed by three expert reviewers, and their assessments together with my own (Reviewing Editor), forms the basis of this letter. We are including the three reviews in their original form at the end of this letter, as there are many specific and useful suggestions in them that will not be repeated in the summary here.

All of the reviewers were impressed with your work. Overall, the experiments appear to be carefully executed. However, there is a general consensus that the data should be more cautiously interpreted and that various experiments that could strengthen (or weaken) a number of the conclusions are feasible.

We would like to encourage you to resubmit a revised manuscript that addresses the specific issues in the reviews. As regards revisions, the reviewers see the molecular part of the work, as opposed to circuit function, is the critical part to focus on. We realize that several of the suggestions are beyond the scope of the present work or what is feasible in a revision, but we have not removed them from the reviews since they convey the reaction to the work over-all. Finally, we hope that you can address the various issues related to clarity; in its present form, the manuscript is quite difficult to read and understand.

*Reviewer #1:*

During eye morphogenesis and retinal development, the roles of Pax6 and its isoforms as well as the mechanisms of Pax6 gene regulation have been extensively studied but are still not completely understood. This manuscript reports the identification of a new LIM domain protein Tgfb1i1 that forms a LIM protein complex with Lhx3 and Isl1 to inhibit the activity of the Pax6 α-enhancer and its auto-binding by Pax6. Site-directed mutagenesis of the α-enhancer, which transcribes the Pax6 isoform Pax6∆PD, leads to loss of GABAergic amacrine cells, whereas both Pax6∆PD overexpression and Tgfb1i1 inactivation, which augments the α-enhancer activity and overproduces Pax6∆PD, results in increased GABAergic amacrine cells in mice. Tgfb1i1 knockout and the α-enhancer mutant mice also display opposite physiological properties: sustained light responses versus more transient ones. This work is a comprehensive in-depth study of the α-enhancer machinery which has revealed a novel function of the little known Pax6∆PD isoform in the specification of a physiological amacrine cell subtype, thus representing a major extension of previous studies on an essential transcriptional regulator of retinal development.

1) The proposed model in Figure 3 is inconsistent with previous reports of the *Lmo4* function and questionable. *Lmo4* is reported to extensively colocalize with Pax6, and its conditional ablation impairs development of both GABAergic and OFF cone bipolar cells whereas its overexpression promotes their differentiation (Duquette et al., PLoS One. 2010 Oct 7;5:e13232; Jin et al., Dev Neurobiol. 2016 Aug;76:900-915). *Lmo4* is thus required for OFF cone bipolar cell specification but the proposed model implies that *Lmo4* is not involved or even has an inhibitory effect on OFF cone bipolar development.

2) There are data that are inconsistent with each other and/or conclusions. For instance, Figure 1 shows Isl1 has no synergistic effect with Pax6 on α-enhancer activation but Figure 1—figure supplement 4B shows a strong effect. Tgfb1i1 is concluded to induce a dose-dependent increase in the association between Lhx3 and Isl1 but the top panel of Figure 1 shows that the interaction between Lhx3 and Isl1 in the absence of Tgfb1i1 is just as strong as in the presence of Tgfb1i1. Also in Figure 1, why did Western blot using anti-flag detect a weak band in all lanes on one panel but not at all on the other two panels? Western with anti-HA had a similar phenomenon. Figure 3 also show inconsistent Vsx2 cell number – the mutant retina has a much thicker layer of Vsx2 cells than that of the control (or are these images in different magnification?). All these raise concerns about data consistency and interpretation.

3) Figure 2—figure supplement 4: Except for some cells co-labeled by Pax6 and GFP, hardly any cells can be seen that are co-stained by other markers and GFP in images shown in A and B. The lack of any examples of convincing colabeled cells raises concerns about the validity of the quantification data in D which show significant increase of Pax6+, Gad67+, calretinin+, and Bhlhb5+(AC) cells and decrease of Vsx2+, G0α+, recoverin+(BP) and Bhlhb5+(BP) cells. Examples of colabeled cells should be displayed at a higher magnification to convince readers their existence. This applies also to Figure 2 where examples of cells colabeled by GABA and GFP should be shown. Another issue is that the authors claim that Pax6 overexpression increased ChAT cells but panel D indicates this is not statistically significant. In addition, what is the scale on images? Is the second image of panel A shown at a higher magnification than others? The ONL, INL and GCL labels on images are too small to be seen and please spell out these abbreviations in the figure legend.

4) Figure 4: No statistical significance is given in panels C and D, which makes it difficult to understand and interpret the data. Also, what does the arrowhead in panel B indicate? Please spell it out in the legend.

*Reviewer #2:*

This paper examines the transcriptional regulation of Pax6 α-enhancer by Lhx3 and Tgfb1i1, and their involvement in the development of the GABAergic amacrine cells in the mouse retina. Pax6 plays a critic role in eye development from the establishment of eye field to the formation of retina and lens. In the retina, Pax6 is expressed throughout retinogenesis and its expression remains in the amacrine and ganglion cells post-development. Prior work in the field has shown that multiple Pax6 transcripts arise from alternative transcriptional initiation and/or alternative splicing. In addition, studies have identified Pax6 α-enhancer, a regulatory region in the intron of Pax6 gene, which contains multiple binding sites for transcription factors including the auto-stimulatory PAX6 binding sites. Nevertheless, how the expression of different Pax6 transcripts is regulated and what are the roles of Pax6 in amacrine subtypes remains unclear.

Earlier work has shown that several additional transcription factors, notably the LIM-homeodomain transcription factors and their negative regulators LMO proteins, are expressed by retinal bipolar, horizontal, amacrine, and ganglion cells. Genetic inactivation of these factors was shown to affect the diversification of retinal neuronal population. Here, Kim et al. report that α-enhancer is specifically activated in GABAergic amacrine cells. By using a proteomic screen, ChIP and in vitro assays, they show that LHX3 and TGFB1I1 interact, bind to the α-enhancer, and inhibit α-enhancer activity in vitro. Tgfb1i1-null retinas display higher α-enhancer activity and generate more GABAergic amacrine cells and fewer OFF bipolar cells. In contrast, removal of Pax6 auto-stimulatory sequences in the α-enhancer in vivo leads to a reduction in Pax6ΔPD and in fewer GABAergic amacrine cells but more OFF bipolar cells. Furthermore, these mutant mice exhibit altered visual acuity and adaptation. Overall, this is a well-performed study and results presented in the paper make a case for the model in which Lhx3 and Tgfb1i1 negatively regulate α-enhancer and the development of GABAergic amacrine cells. However, there are some unresolved issues:

1) This title and manuscript emphasize the involvement of LIM protein, LHX3 in particular, in regulating α-enhancer. There is insufficient data in vivo to support it. The main problem is that GABAergic amacrine cells are mostly born during embryogenesis and peak at E14 (Voinescu et al., 2009, J. Comp. Neurol.). Giving the late-onset of Lhx3 expression at P7 (Elshatory et al., 2007, J. Neurosci.), it is unlikely that Lhx3 regulates the development of GABAergic amacrine cells by regulating α-enhancer. The binding of LHX3 to α-enhancer in vivo (ChIP-PCR results) and LHX3-TGFB1I1 interaction could occur in the bipolar cells, which may or may not be of any functional significance. Experiments such as knocking out or knocking down Lhx3 during amacrine development are desirable to address the role of Lhx3 directly. Also, the authors did a nice CRISPR/Cas9 experiment to demonstrate the necessity of PBS, has a similar experiment done on the LHX3 binding site in DF4?

2) When are the increased GABAergic amacrine cells generated in Tgfb1i1-nulls? This is an important question especially in the light of the late expression of Lhx3. Analysis of GABAergic amacrine cells before Lhx3 expression onset, such as anti-GAD65/67 labeling at P3-5, is needed to determine Tgfb1i1's precise role, and to indirectly support the involvement of Lhx3 in suppressing α-enhancer in bipolar cells – a possible cell fate conversion from bipolar to amacrine cells in Tgfb1i1-nulls.

3) The negative role of LHX3 and TGFB1I1 in the development of GABAergic amacrine cells is an interesting finding. However, their placement in the genetic TF GABAergic amacrine cascade is left to imagination. The story would benefit from trying to analyze how Lhx3 and Tgfb1i1 are related to other known genes of GABAergic amacrine cells such as whether Ptf1 and Bhlhb5 expression is affected in Tgfb1i1-null retinas during amacrine development.

4) The novel finding of GAD67 and P6α-GFP colabeling (Figure 2—figure supplement 2) shows that α-enhancer activity is specific to GABAergic amacrine cells. Interestingly, only a subgroup of GAD67+ cells express P6α-GFP. It is possible that P6α-GFP+ cells represent a unique amacrine subtype. If so, what is the subtype(s) and how is related to the change in visual function seen in Tgfb1i1-nulls? Similarly, is the increase of amacrine cells specific to the INL or GCL or both? It seems like more specific to the INL but it is difficult to tell from the figures unless separate quantification of each layer is done.

5) The visual function tests are done at P60, however, the changes in retinal neurons are assessed at earlier stages. It is somewhat uncertain if there is any additional change in the later stage retinas.

*Reviewer #3:*

Kim et al. investigate the role of the Pax6 α enhancer in retinal neuron fate determination. They find that LIM proteins form a complex (consisting of Isl1, Lhx3, and Tgfb1i1) that inhibits transcription from this enhancer and thereby inhibits amacrine cell fate. The mechanism appears to involve preventing Pax6 itself from binding the α enhancer. Analysis of Tgfb1i1 and Pax6 binding site (deltaPBS) mutant mice suggest that the two proteins may regulate a cell fate choice between GABAergic amacrine and OFF bipolar cell. Behavior and physiology studies demonstrate that amacrine cell number influences sensitivity to light flashes, both in the context of low-threshold detection and light adaptation. I very much like the authors' approach of linking developmental mechanisms to visual function. The molecular studies are thorough and potentially very interesting. However, there are some significant issues with the manuscript in its present form that need to be addressed.

1) The clarity of the manuscript is not sufficient for the reader to understand how the data support the authors' assertions. The major problem here is the structure of the manuscript. Considering the complexity of the voluminous molecular data, there is nowhere near enough explanation. Take for example the panel labeled "Figure 1—figure supplement 6." This is a remarkable amount of work, but the authors do not make it easy for the reader to understand. The figure legend says simply "293T cells were transfected with constructs encoding the indicated protein fragments." Then, in panel L, there is a model figure. Why do the authors feel justified in drawing the model figure this way? I studied the figure for about half an hour, and eventually figured it out. I agree with the authors about their model. But given the format of the e*Life* journal there is no good reason to make the reader do this amount of work for him or herself.

Moreover, this figure is not an isolated case. Another example (one of many I could have selected): In the subsection “Pax6 and LIM protein complex linked by Tgfb1i1 regulate the α-enhancer antagonisticall” the authors write: "Supporting this hypothesis, Lhx3 and Tgfb1i1 antagonized the activation of the α-enhancer in cultured cells (Figure 1; Figure 1—figure supplement 4A-C)." The reader is thereby directed to examine 5 bar graphs that together contain over 70 experiments. These experiments involve not only Lhx3 and Tgfb1i1, but a handful of other molecules (*Lmo4*, Isl1, other Lhx proteins) Which of these many graphs are the ones that support the assertion in the text? The figure legends are again no help. I eventually came to the conclusion that I agree with this particular assertion. But again, given the *ELife* format there is no reason to make things this difficult. I think a reasonable reader might easily decide (at least for the molecular part) that the data do not support the conclusions of this paper – not because the data are incorrect, but because they are not presented in a way that allows the reader to see why the authors came to their conclusion.

To improve clarity the authors could:

A) Use more than four main-text figures, which would allow each figure to have a more focused theme, and would allow incorporation of certain key supplemental data into those figures.

B) Pare away unnecessary data, especially from summary graphs that the authors rely on to make key points highlighted in the text.

C) Write figure legends that highlight key points in complex figures, rather than simply describing what was done.

D) Provide more rationale for why certain experiments were undertaken, both in the Introduction and as the topic sentence in each section of the Results.

2) It's a very compelling idea that a LIM complex comprising Isl1/Lhx3/Tgfb1i1 can regulate a cell fate switch between OFF bipolars and GABAergic amacrine cells. However, I'm unclear on the cellular context in which this is supposed to act. All of these proteins are expressed by postmitotic neurons, mainly in the first postnatal week. So the complex can only exist in neurons. But cell fate decisions are typically made at the progenitor stage, upon cell cycle exit. And GABAergic amacrine cells are typically born before P0 (Cherry et al., 2009, PNAS; Voinescu et al., 2009, J. Comp. Neurol.). So how does the cell fate switch actually work? Is there a transdifferentiation of bipolar cells into amacrines in Tgfb1i1 mutants? Or does the window of GABAergic amacrine cell genesis extend longer, such that they keep getting born when it's supposed to be time for OFF bipolar genesis? Or could the authors have missed the existence of the complex in progenitors? Without a plausible cellular mechanism I remain somewhat skeptical of the notion that this is a simple cell fate switch.

3) In the subsection “Positive correlation between Pax6 α-enhancer-driven Pax6ΔPD expression and GABAergic amacrine cell development” the authors conclude that: "Pax6deltaPD selectively supports GABAergic amacrine cell development but inhibits OFF bipolar cell development." This seems an oversimplification given the data. Full-length Pax6 also increases GABA amacrine cell fate, and suppresses Vsx1 bipolar cell fate. So these features are not selective to the deltaPD version of Pax6. It may not even be true that deltaPD is more efficient than full-length Pax6 at inducing the GABAergic fate, as suggested by Figure 2 (using anti-GABA as a marker), since when Gad67 is used as a marker they appear equally efficient (Figure 2—figure supplement 4).

Also, if the authors wish to claim a selective effect on GABAergic amacrine cells, they should show that glycinergic amacrines are not affected in their Pax6 gain-of-function experiments, and/or in the deltaPBS mutant.

4) In the physiology/behavior section, the model Figure 4 and much of the Results/Discussion focuses on the idea of feedback inhibition to ON or rod bipolar cells. I'm not convinced that these are ON pathway or rod pathway-specific phenotypes. All of the MEA and behavior experiments were done in a light regime where the test stimulus was a brief light stimulus presented in the dark. We don't know what would happen with the inverse paradigm. Even if we did, it is still pretty speculative to get into circuit mechanisms here. The two mutant mice manipulate amacrine cell number in opposite directions. A conservative hypothesis about the physiological effects of these manipulations is that they affect overall inhibitory tone. The interpretations of physiology and behavior experiments probably shouldn't go beyond this.

---

## [Author Response]

*[…] Reviewer #1: […] 1) The proposed model in Figure 3 is inconsistent with previous reports of the Lmo4 function and questionable. Lmo4 is reported to extensively colocalize with Pax6, and its conditional ablation impairs development of both GABAergic and OFF cone bipolar cells whereas its overexpression promotes their differentiation (Duquette et al., PLoS One. 2010 Oct 7;5:e13232; Jin et al., Dev Neurobiol. 2016 Aug;76:900-915). Lmo4 is thus required for OFF cone bipolar cell specification but the proposed model implies that Lmo4 is not involved or even has an inhibitory effect on OFF cone bipolar development.*

As the reviewer indicated, *Lmo4* was suggested to support not only the development of GABAergic amacrine cells but also that of OFF bipolar cells in previous reports. Here, we show Tgfb1i1 inhibits GABAergic amacrine cell fate, although it also supports OFF bipolar cell fate like *Lmo4*. The supportive role of *Lmo4* in the development of both retinal subtypes was proposed based on the reduced numbers of Bhlhb5-positive cells in *Lmo4*-cko mouse retinas (Duquette et al., 2010). Interestingly, the numbers of Bhlhb5-positive type-2 OFF bipolar cell subsets are not significantly changed in Tgfb1i1-KO and Pax6(ΔPBS/ΔPBS) mouse retinas in comparison to WT littermates, whereas Vsx1-positive OFF bipolar cell numbers were significantly altered in those mutant retinas (Figure 4 and Figure 6). The results are also contrary to the changes of Bhlhb5-positive GABAergic amacrine cell subsets, which are increased in Tgfb1i1-ko retinas but are decreased in Pax6(ΔPBS/ΔPBS) retinas. The results therefore suggest that Tgfb1i1 and *Lmo4* might be involved in the development of different OFF bipolar cell subsets, thus the loss of Tgfb1i1 causes the defects in the development of OFF bipolar cell subsets different from those affected in *Lmo4*-cko mice.To avoid the confusion that can be imposed by these complicated relationships (and also to follow the suggestion of other reviewer), we removed the diagram from the revised paper but still wrote in the text (subsection “Tgfb1i1^-/-^ retinas have excessive Pax6 α-enhancer-active GABAergic amacrine cells”, third paragraph).

*2) There are data that are inconsistent with each other and/or conclusions. For instance, Figure 1 shows Isl1 has no synergistic effect with Pax6 on α-enhancer activation but Figure 1—figure supplement 4B shows a strong effect.*

We apologize to present the graph with incomplete data. We missed to normalize the luciferase values by β-gal values. We provide the corrected graph in the revised Figure 2, which shows the synergy.

*Tgfb1i1 is concluded to induce a dose-dependent increase in the association between Lhx3 and Isl1 but the top panel of Figure 1 shows that the interaction between Lhx3 and Isl1 in the absence of Tgfb1i1 is just as strong as in the presence of Tgfb1i1.*

According to the data, Tgfb1i1 likely interferes with the interaction between Isl1 and Lhx3 at low concentration, but it enhances the interaction at high concentration. The opposing effects might be related with the composition of the LIM protein complex. At low concentration, monomeric Tgfb1i1 might prevail and binds Isl1 and Lhx3 separately. Thus, it interferes with the interaction between Isl1 and Lhx3. On the contrary, dimeric (or oligomeric) Tgfb1i1 might be prevalent and forms a heterotetramer with Isl1 and Lhx3, once Tgfb1i1 become abundant (please see Author response image 1). We also modified the expression as “overexpressed Tgfb1i1 further enhanced the association between Lhx3 and Isl1 (subsection “Pax6 and Tgfb1i1 competitively bind Isl1 to antagonistically regulate the α-enhancer”, second paragraph)”.

*Also in Figure 1, why did Western blot using anti-flag detect a weak band in all lanes on one panel but not at all on the other two panels? Western with anti-HA had a similar phenomenon.*

Bands in the second row, which shows Flag-Tgfb1i1 co-immunoprecipitated with HA-Lhx3, are relatively weaker than the others in same figure. We decreased the WB intensity until it does not show non-specific bands overlapping to the size of Flag-Tgfb1i1 (~50 kDa). This also decreased the intensity of specific Tgfb1i1-Flag bands.

*Figure 3 also show inconsistent Vsx2 cell number – the mutant retina has a much thicker layer of Vsx2 cells than that of the control (or are these images in different magnification?). All these raise concerns about data consistency and interpretation.*

We apologize to show an image at higher magnification in that KO sample. We matched the magnification in the revised Figure 6.

*3) Figure 2—figure supplement 4: Except for some cells co-labeled by Pax6 and GFP, hardly any cells can be seen that are co-stained by other markers and GFP in images shown in A and B. The lack of any examples of convincing colabeled cells raises concerns about the validity of the quantification data in D which show significant increase of Pax6+, Gad67+, calretinin+, and Bhlhb5+(AC) cells and decrease of Vsx2+, G0α+, recoverin+(BP) and Bhlhb5+(BP) cells. Examples of colabeled cells should be displayed at a higher magnification to convince readers their existence. This applies also to Figure 2 where examples of cells colabeled by GABA and GFP should be shown.*

In the revised figures, we added the images at higher magnification in the outsets, which can show the co-labeled cells more clearly (Figure 5, Figure 5—figure supplement 2).

*Another issue is that the authors claim that Pax6 overexpression increased ChAT cells but panel D indicates this is not statistically significant.*

We modified the text as “The populations of EGFP-positive cholinergic and glycinergic amacrine cells in pCAGIG-Pax6ΔPD-electroporated mouse retinas are not greatly different from those in pCAGIG-electroporated mouse retinas, but are lower than those in pCAGIG-Pax6-electroporated mouse retinas”.

*In addition, what is the scale on images? Is the second image of panel A shown at a higher magnification than others?*

We provide the information about the scale bar in the figure legend.

*The ONL, INL and GCL labels on images are too small to be seen and please spell out these abbreviations in the figure legend.*

We enlarged the labels and also spelled out the abbreviations in the figure legend.

*4) Figure 4: No statistical significance is given in panels C and D, which makes it difficult to understand and interpret the data.*

We added the statistical significances in the graphs.

*Also, what does the arrowhead in panel B indicate? Please spell it out in the legend.*

The arrowhead points the sustained light-ON response. It is explained in the revised manuscript (subsection “Visual adaptation of the retina is sensitive to Pax6 α-enhancer-active GABAergic amacrine cell number”, second paragraph) and figure legend (Figure 4 has been renumbered to Figure 7 in the revised version).

*Reviewer #2: […] 1) This title and manuscript emphasize the involvement of LIM protein, LHX3 in particular, in regulating α-enhancer. There is insufficient data* in vivo *to support it. The main problem is that GABAergic amacrine cells are mostly born during embryogenesis and peak at E14 (Voinescu et al., 2009, J. Comp. Neurol.). Giving the late-onset of Lhx3 expression at P7 (Elshatory et al., 2007, J. Neurosci.), it is unlikely that Lhx3 regulates the development of GABAergic amacrine cells by regulating α-enhancer. The binding of LHX3 to α-enhancer in vivo (ChIP-PCR results) and LHX3-TGFB1I1 interaction could occur in the bipolar cells, which may or may not be of any functional significance.*

We agree to the reviewer’s opinion that Lhx3 unlikely regulates the development of GABAergic amacrine cells, which are mainly born in the embryonic mouse retina. Here, we propose a model that Lhx3 specifies OFF bipolar cell fate from the post-mitotic neurons, which were born in the embryonic mouse retina and might have default GABAergic amacrine cell fates, in a Tgfb1i1 sensitive manner.

Our subretinal electroporation results show that Pax6ΔPD could support GABAergic amacrine cell fate autonomously (Figure 5; Figure 5—figure supplement 2). It implicates Pax6ΔPD plays a role in generating or maintaining GABAergic amacrine cells in the post-natal mouse retina. Given the lack of newborn GABAergic amacrine cells in the post-natal mouse retina (Figure 6—figure supplement 3; Voinescu et al., (2009) J. Comp. Neurol. 517, 737–750), the electroporation results suggest that Pax6-ΔPD might not support the birth of the cells in the post-natal mouse retina. The Pax6ΔPD might not be important for the development of GABAergic amacrine cells in the embryo either, because the numbers of GABAergic amacrine cells in P4 Pax6(ΔPBS/ΔPBS) mouse retinas are not greatly different from those in WT and Tgfb1i1-KO mouse retinas (Figure 6—figure supplement 3). Therefore, the results suggest that Pax6ΔPD supports the maintenance of the GABAergic amacrine cell fate in the post-natal mouse retina.

On the other hand, Lhx3 and Tgfb1i1 likely contribute to specifying OFF bipolar cell fate, which might be achieved by overcoming default GABAergic amacrine cell fate, via the inhibition of Pax6 α-enhancer-driven Pax6ΔPD expression. In support of this idea, the numbers of OFF bipolar cells, which had incorporated BrdU and exited cell cycle during embryogenesis, were increased in P7 Pax6(ΔPBS/ΔPBS) mouse retinas (Figure 6—figure supplement 3), in which the Pax6ΔPD is decreased by a half of normal level (Figure 6). Conversely, the chance for the fate transition to the OFF bipolar cells was decreased in Tgfb1i1-KO mouse retinas, which express excessive Pax6ΔPD (Figure 5). Based on the opposite changes of GABAergic amacrine and OFF bipolar cell numbers in those two mutant mouse retinas, our results suggest that the BrdU-labeled OFF bipolar cells might be originated from GABAergic amacrine cells, which are not fully differentiated, via transdifferentiation during post-natal days.

Indirect evidence for the transdifferentiation of amacrine cell to bipolar cell has been given by a study traced the fates of Ptf1a-Cre-affected amacrine cells (please see the presence of Chx10;R26R double-positive bipolar cells in Figure 3 of Fujitani et al. (2006) Development 133, 4439-50). However, it is unclear whether those bipolar cells derived from Ptf1a-Cre-active embryonic amacrine cells are OFF types. The direct conversion of GABAergic amacrine cells to OFF bipolar cells and the inactivation of Pax6 α-enhancer in this process should be confirmed in future study by tracing fates of individual cell in the post-natal mouse retina.

*Experiments such as knocking out or knocking down Lhx3 during amacrine development are desirable to address the role of Lhx3 directly.*

Lhx3-KO mice are perinatally lethal (Sheng et al. (1996) Science 272, 1004-7), thus we could not have a chance to analyze the post-natal retinal phenotypes. Lhx3-flox mice, which can be useful for deleting Lhx3 gene in the embryonic retina where amacrine cells develop, are not available, yet. We, thus, knocked Lhx3 gene out by CRISPR/Cas9 using subretinal electroporation of two independent sgRNAs against Lhx3 to P0 mouse retinas.

We found that cells expressing Lhx3 sgRNA and Cas9 (indirectly labeled by GFP expressed from co-electroporated pCAGIG construct) were strongly enriched in the bottom half of INL where amacrine cells are enriched, whereas the cells express only Cas9 were enriched in the top half of INL where bipolar and Muller glial cells are enriched (Figure 4—figure supplement 2). Moreover, significant numbers of those Lhx3 sgRNA-expressing (i.e., GFP-positive) cells are positive to GABAergic amacrine cell markers (Gad67, GABA, and Bhlhb5), whereas only few of Cas9-only control cells express those markers (Figure 4—figure supplement 2). On the contrary, almost none of Lhx3 sgRNA-expressing cell expresses OFF bipolar cell marker Vsx1 (Figure 4—figure supplement 2). Overall, the results are consistent with those of Tgfb1i1-KO mice (Figure 4). Therefore, as we explain in the above, the results implicate that Lhx3 and Tgfb1i1 might specify OFF bipolar cell fate by suppressing default GABAergic amacrine cell fate, which is supported by Pax6ΔPD, in the post-natal mouse retina.

*Also, the authors did a nice CRISPR/Cas9 experiment to demonstrate the necessity of PBS, has a similar experiment done on the LHX3 binding site in DF4?*

We had also tried to delete DF3 and DF4 by CRISPR/Cas9 system, but we failed to obtain the mice with deletions in those sequences. We think deletion(s) of off-target genes by those DF3 and DF4 sgRNAs might cause the lethality, as their off-target probabilities were higher than those for the PBS sgRNAs.

*2) When are the increased GABAergic amacrine cells generated in Tgfb1i1-nulls? This is an important question especially in the light of the late expression of Lhx3. Analysis of GABAergic amacrine cells before Lhx3 expression onset, such as anti-GAD65/67 labeling at P3-5, is needed to determine Tgfb1i1's precise role, and to indirectly support the involvement of Lhx3 in suppressing α-enhancer in bipolar cells – a possible cell fate conversion from bipolar to amacrine cells in Tgfb1i1-nulls.*

By following the reviewer’s suggestion, we analyzed with P4 Tgfb1i1-KO and Pax6(ΔPBS/ΔPBS) mouse retinas, and found the numbers of GABAergic amacrine cell are not changed significantly in comparison to WT littermates (Figure 6—figure supplement 3). The results, together with other results in Figure 6—figure supplement 3, implicate that Tgfb1i1 may not suppress the production of GABAergic amacrine cells but may negatively affect the maintenance of the cells (please see our first answer to the reviewer’s comment 1).

*3) The negative role of LHX3 and TGFB1I1 in the development of GABAergic amacrine cells is an interesting finding. However, their placement in the genetic TF GABAergic amacrine cascade is left to imagination. The story would benefit from trying to analyze how Lhx3 and Tgfb1i1 are related to other known genes of GABAergic amacrine cells such as whether Ptf1 and Bhlhb5 expression is affected in Tgfb1i1-null retinas during amacrine development.*

Ptf1a is not expressed in mature mouse retina, but it is expressed in the embryonic mouse retina and play important role in the development of amacrine cells (Fujitani et al. (2006) Development 133, 4439-50). Thus, we examined the distribution of Bhlhb5-positive cells in the mature retinas. We found the numbers of Bhlhb5-positive GABAergic amacrine cell subsets were increased in P14 Tgfb1i1-KO mouse retinas (Figure 4). Moreover, those include Pax6 α-enhancer active cell population (Figure 4—figure supplement 1). However, it is unclear whether Lhx3 and Tgfb1i1 regulate Bhlhb5 expression directly or indirectly via Pax6 α-enhancer inhibition. The mechanism should be studied separately.

*4) The novel finding of GAD67 and P6α-GFP colabeling (Figure 2—figure supplement 2) shows that α-enhancer activity is specific to GABAergic amacrine cells. Interestingly, only a subgroup of GAD67+ cells express P6α-GFP. It is possible that P6α-GFP+ cells represent a unique amacrine subtype. If so, what is the subtype(s) and how is related to the change in visual function seen in Tgfb1i1-nulls?*

Unfortunately, we currently have no clear idea about the molecular and electrophysiological identities of Pax6 α-GFP-positive GABAergic amacrine cells. The molecular and electrophysiological identifications of P6α GABAergic amacrine cells should be left for our future study.

*Similarly, is the increase of amacrine cells specific to the INL or GCL or both? It seems like more specific to the INL but it is difficult to tell from the figures unless separate quantification of each layer is done.*

We counted the numbers of Pax6 α-GFP-positive cells in INL and GCL separately, and provide the results for the reviewers’ inspection only due to the limited space in the figure (please see Author response image 2). The results indicate that the changes are observed mainly in INL of the mouse retinas.

*5) The visual function tests are done at P60, however, the changes in retinal neurons are assessed at earlier stages. It is somewhat uncertain if there is any additional change in the later stage retinas.*

We provide the retinal phenotypes of P60 mice in Figure 7—figure supplement 2. Overall, the results are not significantly different from those at P14.

*Reviewer #3:*

*[…] 1) The clarity of the manuscript is not sufficient for the reader to understand how the data support the authors' assertions. The major problem here is the structure of the manuscript. Considering the complexity of the voluminous molecular data, there is nowhere near enough explanation. Take for example the panel labeled "Figure 1—figure supplement 6." This is a remarkable amount of work, but the authors do not make it easy for the reader to understand. The figure legend says simply "293T cells were transfected with constructs encoding the indicated protein fragments." Then, in panel L, there is a model figure. Why do the authors feel justified in drawing the model figure this way? I studied the figure for about half an hour, and eventually figured it out. I agree with the authors about their model. But given the format of the eLife journal there is no good reason to make the reader do this amount of work for him or herself.*

*Moreover, this figure is not an isolated case. Another example (one of many I could have selected): In the subsection “Pax6 and LIM protein complex linked by Tgfb1i1 regulate the α-enhancer antagonisticall” the authors write: "Supporting this hypothesis, Lhx3 and Tgfb1i1 antagonized the activation of the α-enhancer in cultured cells (Figure 1; Figure 1—figure supplement 4A-C)." The reader is thereby directed to examine 5 bar graphs that together contain over 70 experiments. These experiments involve not only Lhx3 and Tgfb1i1, but a handful of other molecules (Lmo4, Isl1, other Lhx proteins) Which of these many graphs are the ones that support the assertion in the text? The figure legends are again no help. I eventually came to the conclusion that I agree with this particular assertion. But again, given the eLife format there is no reason to make things this difficult. I think a reasonable reader might easily decide (at least for the molecular part) that the data do not support the conclusions of this paper – not because the data are incorrect, but because they are not presented in a way that allows the reader to see why the authors came to their conclusion.*

*To improve clarity the authors could:*

*A) Use more than four main-text figures, which would allow each figure to have a more focused theme, and would allow incorporation of certain key supplemental data into those figures.*

We apologize for the inconvenience by making our paper too compressed. We have reformatted the paper with 7 main figures. We also provide the details of the results in the revised Figure 2 and Figure 3 following the reviewer’s suggestion.

*B) Pare away unnecessary data, especially from summary graphs that the authors rely on to make key points highlighted in the text.*

We removed summary diagrams in original Figure 1 and Figure 3. We also made the revised Figure 3 more simplified and clearer than the original version.

*C) Write figure legends that highlight key points in complex figures, rather than simply describing what was done.*

We added key points and experimental descriptions in the revised figure legends, in addition to the explanations in main text.

*D) Provide more rationale for why certain experiments were undertaken, both in the Introduction and as the topic sentence in each section of the Results.*

We provided the rationales for the experiments in the revised manuscript.

*2) It's a very compelling idea that a LIM complex comprising Isl1/Lhx3/Tgfb1i1 can regulate a cell fate switch between OFF bipolars and GABAergic amacrine cells. However, I'm unclear on the cellular context in which this is supposed to act. All of these proteins are expressed by postmitotic neurons, mainly in the first postnatal week. So the complex can only exist in neurons. But cell fate decisions are typically made at the progenitor stage, upon cell cycle exit. And GABAergic amacrine cells are typically born before P0 (Cherry et al., 2009, PNAS; Voinescu et al., 2009, J. Comp. Neurol.). So how does the cell fate switch actually work? Is there a transdifferentiation of bipolar cells into amacrines in Tgfb1i1 mutants? Or does the window of GABAergic amacrine cell genesis extend longer, such that they keep getting born when it's supposed to be time for OFF bipolar genesis? Or could the authors have missed the existence of the complex in progenitors? Without a plausible cellular mechanism I remain somewhat skeptical of the notion that this is a simple cell fate switch.*

According to the lack of expression in the embryonic mouse retina, the LIM protein complex may start to act in the post-natal mouse retina. In addition, the opposite changes in the numbers of GABAergic amacrine cells and OFF bipolar cells in Tgfb1i1-KO and Pax6(ΔPBS/ΔPBS) mouse retinas suggest that the fates of these two cell types might be switchable. Combining these, we hypothesize that the LIM complex likely contributes to specifying OFF bipolar cell fate, which might be achieved by overcoming default GABAergic amacrine cell fate in the post-natal mouse retina, via the inhibition of Pax6 α-enhancer-driven Pax6-ΔPD expression.

Our subretinal electroporation results show that Pax6ΔPD could support GABAergic amacrine cell fate autonomously (Figure 5; Figure 5—figure supplement 2). It implicates Pax6ΔPD plays a role in generating or maintaining GABAergic amacrine cells in the post-natal mouse retina. Given the lack of newborn GABAergic amacrine cells in the post-natal mouse retina (Figure 6—figure supplement 3; Voinescu et al., (2009) J. Comp. Neurol. 517, 737–750), the electroporation results suggest that Pax6-ΔPD might not support the birth of the cells in the post-natal mouse retina. The Pax6ΔPD might not be important for the development of GABAergic amacrine cells in the embryo either, because the numbers of GABAergic amacrine cells in P4 Pax6(ΔPBS/ΔPBS) mouse retinas are not greatly different from those in WT and Tgfb1i1-KO mouse retinas (Figure 6—figure supplement 3). Therefore, the results suggest that Pax6ΔPD supports the maintenance of the GABAergic amacrine cell fate in the post-natal mouse retina.

On the other hand, Lhx3 and Tgfb1i1 likely contribute to specifying OFF bipolar cell fate, which might be achieved by overcoming default GABAergic amacrine cell fate, via the inhibition of Pax6 α-enhancer-driven Pax6ΔPD expression. In support of this idea, the numbers of OFF bipolar cells, which had incorporated BrdU and exited cell cycle during embryogenesis, were increased in P7 Pax6(ΔPBS/ΔPBS) mouse retinas (Figure 6—figure supplement 3), in which the Pax6ΔPD is decreased by a half of normal level (Figure 6). Conversely, the chance for the fate transition to the OFF bipolar cells was decreased in Tgfb1i1-KO mouse retinas, which express excessive Pax6ΔPD (Figure 5). Based on the opposite changes of GABAergic amacrine and OFF bipolar cell numbers in those two mutant mouse retinas, our results suggest that the BrdU-labeled OFF bipolar cells might be originated from GABAergic amacrine cells, which are not fully differentiated, via transdifferentiation during post-natal days.

Indirect evidence for the transdifferentiation of amacrine cell to bipolar cell has been given by a study traced the fates of Ptf1a-Cre-affected amacrine cells (please see the presence of Chx10;R26R double-positive bipolar cells in Figure 3 of Fujitani et al. (2006) Development 133, 4439-50). However, it is unclear whether those bipolar cells derived from Ptf1a-Cre-active embryonic amacrine cells are OFF types. The direct conversion of GABAergic amacrine cells to OFF bipolar cells and the inactivation of Pax6 α-enhancer in this process should be confirmed in future study by tracing fates of individual cell in the post-natal mouse retina.

*3) In the subsection “Positive correlation between Pax6 α-enhancer-driven Pax6ΔPD expression and GABAergic amacrine cell development” the authors conclude that: "Pax6deltaPD selectively supports GABAergic amacrine cell development but inhibits OFF bipolar cell development." This seems an oversimplification given the data. Full-length Pax6 also increases GABA amacrine cell fate, and suppresses Vsx1 bipolar cell fate. So these features are not selective to the deltaPD version of Pax6. It may not even be true that deltaPD is more efficient than full-length Pax6 at inducing the GABAergic fate, as suggested by Figure 2 (using anti-GABA as a marker), since when Gad67 is used as a marker they appear equally efficient (Figure 2—figure supplement 4).*

Gad67;EGFP double-positive GABAergic amacrine cell population (44%) is about a half of Syntaxin;EGFP double-positive pan-amacrine population (85%) in pCAGIG-Pax6-electroporated retinas (Figure 5—figure supplement 2). The ratio in pCAGIG-electroporated retinas was also not quite different (15% (Gad67;EGFP) vs. 27% (Syntaxin;EGFP)) from that. In those two groups, Gad67;EGFP double-positive GABAergic amacrine cell population are also not greatly different from GlyT1;EGFP double-positive glycinergic amacrine cell population. In contrast, Gad67;EGFP double-positive GABAergic amacrine cell population (46% ± 7.33%) is as big as Syntaxin;GFP double-positive amacrine cell population (54% ± 7.56%) in pCAGIG-Pax6ΔPD-electroporated retinas. Those GABAergic amacrine cell population is over 5-folds of GlyT1;EGFP double-positive cell population (8% ± 1.6%). Therefore, the results could support the idea that GABAergic amacrine cell subtypes were produced preferentially by Pax6ΔPD overexpression.

To help our readers’ understanding, we rewrote as “Moreover, by showing insignificantly different marker positivity with EGFP;Syntaxin double-positive INL cells (54% ± 7.56%(Syntaxin) vs. 46% ± 7.33%(Gad67)), majority of EGFP-positive amacrine cells in the pCAGIG-Pax6ΔPD-electroporated retinas are predicted as GABAergic amacrine cells, which are approximately half of the EGFP;Syntaxin double-positive amacrine cell population in pCAGIG-Pax6-electroporated mouse retinas (85% ± 7.2%(Syntaxin) vs. 44% ± 9.17%(Gad67)) (Figure 5; Figure 5—figure supplement 2 [second row], D)”. We also rewrote the conclusion sentence as “…Pax6ΔPD preferentially supports GABAergic amacrine cell fate, while full-length Pax6 induces all amacrine cell types in a similar ratio observed in normal mouse retina”.

*Also, if the authors wish to claim a selective effect on GABAergic amacrine cells, they should show that glycinergic amacrines are not affected in their Pax6 gain-of-function experiments, and/or in the deltaPBS mutant.*

We added GlyT1 immunostaining results in Figure 5—figure supplement 2. The results suggest that Pax6ΔPD did not increase the number of GlyT1-positive cells significantly.

*4) In the physiology/behavior section, the model Figure 4 and much of the Results/Discussion focuses on the idea of feedback inhibition to ON or rod bipolar cells. I'm not convinced that these are ON pathway or rod pathway-specific phenotypes. All of the MEA and behavior experiments were done in a light regime where the test stimulus was a brief light stimulus presented in the dark. We don't know what would happen with the inverse paradigm. Even if we did, it is still pretty speculative to get into circuit mechanisms here. The two mutant mice manipulate amacrine cell number in opposite directions. A conservative hypothesis about the physiological effects of these manipulations is that they affect overall inhibitory tone. The interpretations of physiology and behavior experiments probably shouldn't go beyond this.*

We agree that the experiments in this paper are not enough to get into circuit information and additional measurements in various light conditions should be done to reach more accurate conclusion. Following the reviewer’s suggestion, we modified the text and diagram by removing the feedback component and simply emphasizing the alteration of inhibitory tone in light-ON pathway in those two mutant mouse retinas.